# MENTALBLACKBOARD : EVALUATING SPATIAL VISUALIZATION VIA MATHEMATICAL TRANSFORMATIONS

## ABSTRACT

Spatial visualization is the mental ability to imagine, transform, and manipulate the spatial characteristics of objects and actions. This intelligence is a part of human cognition where actions and perception are connected on a mental level. Do state-of-the-art Vision-Language Models (VLMs) also exhibit this ability? To explore this, we develop MentalBlackboard, an open-ended spatial visualization benchmark for Paper Folding and Hole Punching tests within two core tasks: prediction and planning. Our prediction experiments reveal that models mostly overpredict the final hole numbers and struggle with applying symmetrical transformations, even when they predict the sequence of unfolding steps correctly. The backward folding process (folding the paper away from the camera/observer), which leads to limited vision, reduces the accuracy of spatial arrangement construction for certain models. Rotations, which alter the orientation of the unfolding actions, introduce a significant challenge for models to understand the physical orientation of the paper. The planning task, in which models are required to identify the sequence of folds that match the final hole pattern, shows models' limitations in analyzing symmetrical relations and creating the multi-stage symmetry process. In the task of generalization, which does not require spatial visualization, models reason through the visual analogies involving two visual examples of the same paper-folding process, along with a distinct spatial property and text-based hole information. Although the best-performing model, o3, achieves a peak performance of 71.6% in transferring spatial data, it only obtains 25% accuracy on text-based prediction tasks. Claude Opus 4.1 achieves the highest planning score with 10%. The field-wise performance shows that models struggle more with locating and orienting the holes.

## 1 INTRODUCTION

Spatial thinking is an essential component of nonverbal intelligence, allowing the brain to perform cognitive tasks such as imagining, altering structures, and shifting perspectives. This ability is directly connected with the success of the Science, Technology, Engineering, and Mathematics (STEM) fields (Wai et al., 2009). Visualizing the 2D structure as a 3D transformation in space is a required tool to mentally design new systems. Engineers typically utilize a highly developed spatial visualization capability (Maresch & Sorby, 2021) to solve spatial-geometric problems. Similarly, in mathematics, spatial reasoning is employed to comprehend space, visualize objects and their associations, and tackle problems (Woolcott, 2020). This skill is critical for grasping the complicated and abstract concepts in mathematics (Clements & Battista, 1992; Pirie & Kieren, 1994).

Spatial visualization represents one of the object-centered spatial abilities, which involves intricate spatial manipulations within the object's borders (Harris et al., 2023). The *Paper Folding Test (PFT)* (Ekstrom et al., 1976), one of the multi-stage spatial visualization tasks, involves a series of paper-folding actions followed by a hole-punching step. It challenges participants to identify the spatial arrangements of the resulting holes after mentally unfolding the paper. This mental folding task is connected to multiplicative and algebraic processes (Empson & Turner, 2006) and inherently contains symmetry transformation (Uttal et al., 2024). While reflections across the crease line are connected with spatial relations, the final appearance of holes depicts the structural transformation (Harris & Lowrie, 2024).

Although spatial visualization plays a significant role in mathematical thinking (Medina Herrera et al., 2024), its evaluation in state-of-the-art vision-language models (VLMs) remains relatively underexplored, especially compared to other spatial reasoning tasks such as spatial perception (Hu et al., 2022), spatial relations (Wiebrock et al., 2000), and navigation & mapping (Gupta et al., 2017). Many spatial benchmarks often rely on a multiple-choice format (Zeno et al., 2025; Yang et al., 2025b) that requires selecting the correct answer from a provided solution set. However, this approach reveals only accuracy and does not elucidate the reasons for incorrect responses. In addition, relying on limited answer options can fail to evaluate the underlying reasoning abilities; rather, it could expose alternative

strategies to solve the problem such as elimination (discarding options that do not match the stimulus features), perceptual matching (scanning the similarities between options and stimulus components) or working backward (testing each option to find the best fit) (Snow, 1980).

In response to these challenges, we introduce MentalBlackboard: an open-ended spatial visualization benchmark that employs PFT within two core tasks: prediction and planning. Prediction integrates PFT to rotation transformations and implements a folding-unfolding strategy for the solution. This approach requires mental transformation steps to identify the hole configurations, which demands a high cognitive load to measure the visualization process Harris et al. (2013), Preuss et al. (2024). The planning task aims to interpret the final unfolded paper and determine the folds and initial holes to reach the identical result. Both tasks require visual perception to understand the concept, visuospatial working memory to track multiple folds and punches while processing, sequential reasoning to understand the ordered folds/unfolds, and spatial visualization to imagine and manipulate objects. Task examples are shown in Figure 2.

Our contributions and observations are summarized below:

- We introduce MentalBlackboard, a large-scale, open-ended benchmark to evaluate VLMs' spatial visualization skills with prediction and planning tasks. We develop an automated data creation pipeline to dynamically generate the 3D animation of the tasks.
- We evaluate state-of-the-art VLMs on MentalBlackboard tasks, showing up to 25% accuracy in text-based prediction and 10% planning tasks. We identify the main challenges through the analysis of the open-ended evaluation results.
- We conduct additional ablation studies to assess spatial information transfer and the impact of backward folding on prediction performance.

## 2 RELATED WORK

**Spatial Visualization.** Spatial visualization ability requires multi-stage mental processes, which makes it a complex task to perform. It includes various tasks in which mathematical transformations are applied in different way (Harris et al., 2023). While PFT (Ekstrom et al., 1976) implements symmetrical processes and manipulation of the object, the Mental Rotation Task (MRT) (Shepard & Metzler, 1971) focuses on rotation without transforming the object, which does not fulfill the definition of visualization (Fehringer, 2020); therefore, it is categorized between spatial visualization and spatial relations (Pellegrino et al., 1984). Another 2D to 3D conversion task similar to PFT is the Surface Development Test (Ekstrom et al., 1976), which involves imagining how a flat shape folds into a 3D object but does not require tracking the spatial components. To assess the intelligence of visualization, these pen-and-paper tasks have been typically utilized. Recent spatial visualization benchmarks, such as Ramakrishnan et al. (2025), Jia et al. (2025), Li et al. (2025b), and Stogiannidis et al. (2025), address some of these tasks with valuable but limited perspectives. These benchmarks contain multiple spatial reasoning tasks, including spatial visualization (SV). Each SV task relies on a distinct underlying transformation: MRT emphasizes mental rotation, whereas PFT centers on mirror symmetry. The memory use and tracking of actions also show differences for each task. In contrast, the MentalBlackboard dataset focuses on one task (PFT), which was revised to include both symmetry and rotation transformations, along with a memory requirement. Beyond the extra mental process, this task differs from the other PFT tasks in Table 1 in terms of its large scale, multi-dimensional structure, and open-ended evaluation methodology, which enables exploring the reasons behind the failure cases. Unlike Mind the Gap, which restricts tasks to at most two folds and prohibits combining diagonal with other fold types (Stogiannidis et al., 2025), MentalBlackboard introduces high cognitive complexity by supporting up to four folds (excluding rotation), and generating 12K unique configurations with an automated pipeline. This complexity arises from the increased number of combinations of folds, especially those involving diagonal folds, which lead to asymmetry, occlusion, and abstract non-perceptual matching (Kyllonen et al., 1984; Burte et al., 2019).

**Spatial Reasoning Benchmarks.** Spatial reasoning requires the understanding and inference from the spatial data, actions, and their relations in space. It contains diverse tasks that measure distinct properties of the spatial reasoning process. While benchmarks like Visual Spatial Reasoning (VSR) (Liu et al., 2023), Clevr (Johnson et al., 2016), SpatialSense (Yang et al., 2019), SpatialVLM (Chen et al., 2024) and VSI-Bench (Yang et al., 2024) focus on relations between spatial arrangements in diverse environments, GOAT-Bench (Harris et al., 2023), Navi2Gaze (Zhu et al., 2024), and VLABench (Zhang et al., 2024) target spatial orientation and navigation. Several datasets, such as ALFRED (Shridhar et al., 2020), AGQA (Grunde-McLaughlin et al., 2021), and NExT-QA (Xiao et al., 2021), address spatio-temporal tasks, where models need to reason about both spatial relationships and temporal dynamics across sequences of events or visual inputs. However, MentalBlackboard differentiates itself by focusing on multi-stage mental manipulations utilizing symmetry and rotation transformations. It challenges models to track the actions, to understand the physical dynamics of the object, and to perform a series of mental actions. This high cognitive load

positions our proposed MentalBlackboard as a distinct benchmark for evaluating the spatial visualization ability of the VLMs.

**Vision-Language Models** Vision-Language Models (VLMs) have made significant progress across perceptual understanding, reasoning, and language tasks. The integration of effective linguistic ability and visual perception enables the performance of multimodal tasks such as visual question answering (Goyal et al., 2017; Marino et al., 2019), image captioning (Chen et al., 2015; Sharma et al., 2018; Plummer et al., 2015), visual grounding (Yu et al., 2016), and visual reasoning (Suhr et al., 2019; Hudson & Manning, 2019). This multimodality increases interest in assessing the VLM's capability in complex reasoning tasks. The MentalBlackboard pushes the boundaries of the models' spatial reasoning intelligence, mainly focusing on the multi-stage manipulation in the 2D to 3D conversion task.

Table 1: Comparison of spatial reasoning datasets and their SV tasks on cognitive processes, size, space, and evaluation approach. Meanings of the abbreviations: Symmetry = Does the task involve mental transformation based on symmetry?, Memory = Does the task in datasets require tracking actions?, Rotation = Does mental rotation exhibit in the task?, MC = multiple-choice, MPFB = Minnesota Paper Form Board, SBST = Santa Barbara Solid Test Cohen & Hegarty (2012), R-Cub-Vis = R-Cube Visualization Short Test, CUR-MA = Cube Unfolding and Reconstruction and Mental Animation, VP-MRT = Visual Penetration and Mental Rotation Task. If a dataset includes multiple SV tasks that obtain different mental operations, their results are concatenated with a plus symbol.

| | Symmetry | Memory | Rotation | Size | Dimension | Evaluation | SV Tasks |
|---|---|---|---|---|---|---|---|
| SPACE (Ramakrishnan et al., 2025) | ✗ | ✗ | ✓ | 172/50 (visual) | 2D | MC | MRT + MPFB (Likert & Quasha, 1941) |
| STARE (Li et al., 2025b) | ✗ | ✗ | ✓ | 320 | 2D | Yes/No | Cube Folding |
| Mind the Gap (Stogiannidis et al., 2025) | ✓+✗ | ✓+✗ | ✗+✓ | 200/400 | 2D | MC | PFT + MRT |
| BSA (Xu et al., 2025) | ✗ | ✗+✓ | ✓ | 90 | 2D | MC | SBST + R-Cube-Vis (Fehringer, 2020) |
| Omni Spatial (Jia et al., 2025) | ✓+✗ | ✓+✗ | ✗+✓ | < 245 | 2D | MC | PFT + MRT |
| SpatialViz-Bench (Wang et al., 2025a) | ✓+✗+✗ | ✓+✗+✓ | ✗+✓+✓ | 1180 | 2D | MC | PFT + VP-MRT + CUR-MA |
| MentalBlackboard (ours) | ✓ | ✓ | ✓ | > 1M | 2D, 3D | Open-ended | PFT (included rotation) |

## 3 MENTALBLACKBOARD

The MentalBlackboard benchmark is developed to evaluate the spatial visualization intelligence of state-of-the-art large language models across two core tasks: prediction and planning. This benchmark challenges models to process the sequence of visual actions or the results of actions and apply the symmetry and rotation transformations to the three-dimensional object multiple times, regarding the task. The implementation of an open-ended evaluation framework provides both accuracy and error analysis for the failure cases. The dynamic structure of the creation process enables a large-scale dataset of over 12,000 unique configurations without implementing rotations. Figure 1 displays the dataset creation pipeline of MentalBlackboard.

### 3.1 BENCHMARK CONSTRUCTION

**Environment Setup.** The Blackboard dataset employs a paper-folding and hole-punching test structure. To develop a physically dynamic system that implements both symmetry and rotation in a three-dimensional environment, we utilized the VPython platform. However, building the paper structure and applying the various creasing points is challenging due to the physical limitations of the objects in the environment. To execute diagonal foldings without any physical violations, we divide the paper into 32 triangles, which limits the number of possible folding steps to five.

**Dataset Configuration.** The Blackboard benchmark is generated utilizing the PFT approach through a series of folding actions followed by a hole punching process. We identify eight fold types: horizontal, vertical, diagonal, and their directions, and three rotation angles: 90, 180, and 270 degrees. The spatial attributes of each hole include its location, orientation, geometric shape, and size. To generate the hole to be punched in the paper, we select a combination of nine distinct shapes, two size options, four orientations, and 32 defined positions.

In this benchmark, each problem begins with a square flat paper that presents a 2D plane within 3D space, featuring two sides: front and back. The crease line for each fold is computed by dividing the region in half in accordance with the folding type, ensuring a symmetric folding process. To eliminate physically infeasible folding sequences, multiple validation rules are introduced, which preserve the integrity of the paper mesh during folding; see Appendix D for details. Once the initial fold is completed, the part of the paper transitions to a folded state, and the next folding step is applied to the remaining paper regions. The folded segments can be stacked above or below the rest of the paper,

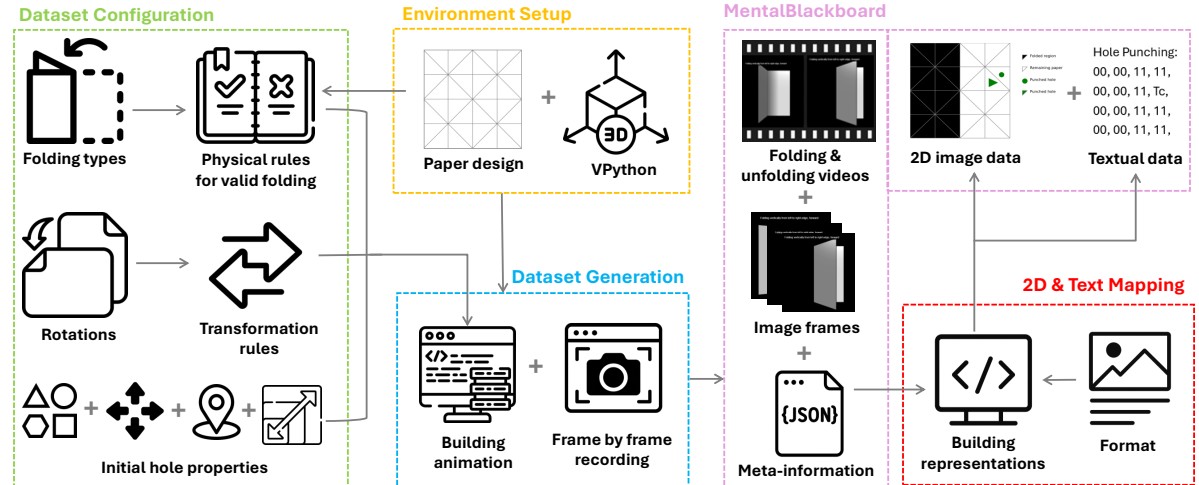

Figure 1: The MentalBlackboard creation pipeline, which consists of environment setup, dataset configuration, dataset generation, and 2D and text mapping.

depending on whether the fold is *forward* (toward the camera/observer) or *backward* (away from the camera/observer). The rotation is applied after the folding action(s), which reorients prior folds by altering the direction of the crease line. To analyze the physical transformations that occur after rotation, we generated transformation rules that define the impact of the rotation angle on the type and direction of folding (see Appendix D for details). During unfolding, these rules provide physical awareness of the oriented paper and ensure that the paper mesh does not experience physical deformation or self-intersection.

**Dataset Generation.** To construct the 3D folding paper animations, predetermined folding and rotation types are combined, and the combinations are validated to determine whether the structure is physically applicable to the paper design. A total of nine unique folding step configurations are generated, five of which include rotation (see Appendix C). To determine the initial hole data for the punching action, we randomly select the size, location, shape, and direction information from the identified sets. To animate the unfolding actions, we apply the reverse direction and order of the folding actions. However, if the task implements rotation, the transformation rules (see Appendix D) are applied to the paper to identify the physical orientation of the paper layout and alteration of the creasing line. VPython renders animations in-browser; therefore, each video frame is captured via automated screenshots and then converted into videos. Besides folding & unfolding frames and videos, the metadata of each task has been saved in a JSON file.

**2D & Text Mapping.** To observe the performance difference between diverse modalities, we design a 2D image and a textual representation of the folding, punching, and unfolding processes. The image format utilizes the identical paper design with animation and employs colors, black, white, and green, to depict the folded part, the remaining region of paper, and punched holes, respectively. The text-based format utilizes symbols: 0, 1, and encoding letters to represent folded, unfolded, and hole shapes. Because of the limitations of letter symbols, direction information is not applied to the text mapping. Unlike other formats, text mapping implements a grid structure, and each cell consists of two values: left and right triangles. The structure of textual expression is depicted as [row,column,tri].

### 3.2 TASKS OF MENTALBLACKBOARD

**Prediction.** The prediction task tests the models' reasoning and implementation process about how to reach the result when initial data is provided. The sequence of folding actions, the numbered paper design, and initial hole information (shape, size, direction, and location), are provided with the model. The task aims to understand the folding actions, reason about how to unfold the paper by applying symmetry-related transformations, and find the resulting holes. The models need to consider the physical effect of the rotation transformations and decide the unfolding steps accordingly. The task is designed with three spatial encodings: text, 2D image, and video. The example prediction task with distinct representations is shown in Figure 2.

**Planning.** This task evaluates whether the model can reach the goal by reasoning through multiple steps. The final unfolded paper image with textual hole information is provided to the model, and the model is requested to find the

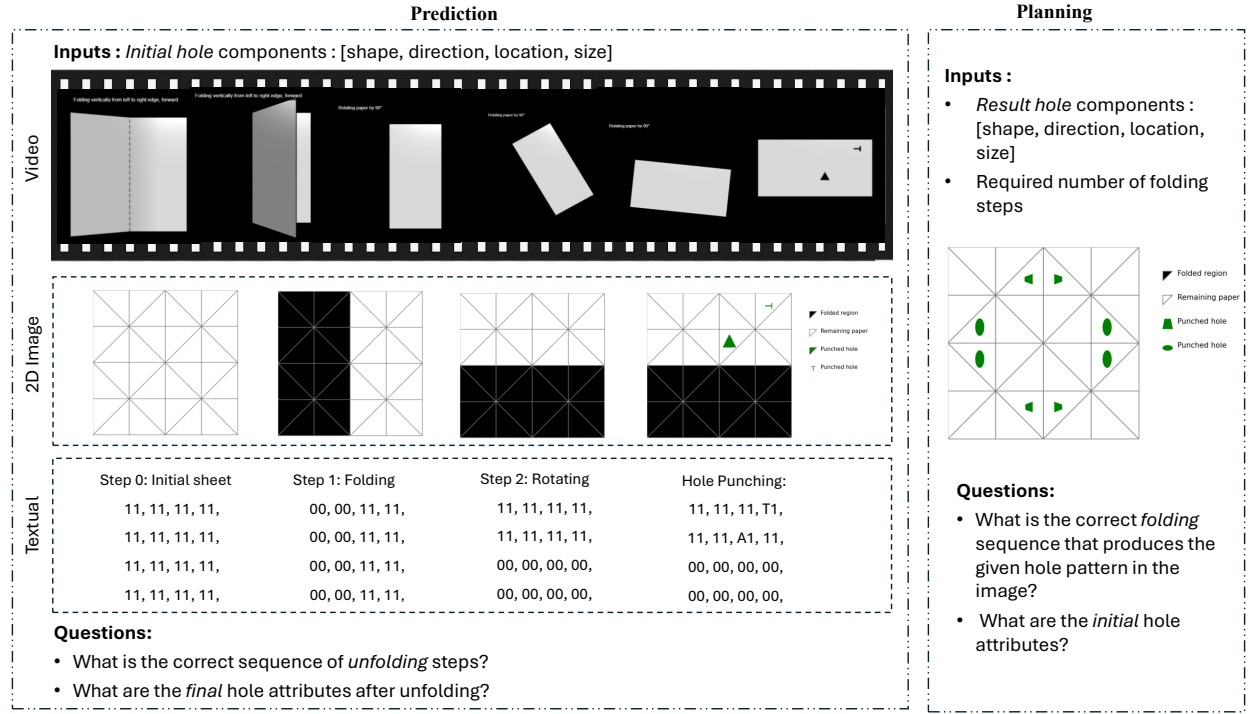

Figure 2: Examples of a prediction task and a planning task.

sequence of folding actions and initial hole information. To prevent models from selecting trivial one-step solutions, we also provide the required number of folding steps to reach the final result. This task does not include rotation actions and limits the maximum number of initial holes to two. Figure 2 also shows a planning task example.

## 4 EVALUATION SETUP

**MentalBlackBoard Test Dataset.** To evaluate spatial visualization in the prediction task, we generated three types of representations: (1) video (MP4 or image frames), (2) 2D abstract images, and (3) text descriptions designed for large language models (LLMs). A total of 900 prediction problems were created for the video format and 315 for the other formats. The task consists of 9 distinct configurations, with 5 involving rotation (see Appendix C). The statistics of the prediction task in terms of used fold types, position of the initial holes, and shape of the holes are illustrated in Figure 3. The unfolding types are identical to the fold types. On the other hand, the planning task utilizes a 2D image representation and includes 400 questions applied in 4 distinct structures, *excluding rotation*. In both tasks, only forward folding (toward the camera/observer) is applied.

**Benchmark Models.** We evaluate open-sourced and proprietary models. For video-based prediction, we evaluated Claude Opus 4.1 (Anthropic, 2025b), o3 (OpenAI, 2025b), GPT-5.1 (OpenAI, 2025a) LLaVa-OneVision (Li et al., 2025a), and Qwen-VL 2.5 (7B) (Bai et al., 2025), and combined them with Claude 4 (Anthropic, 2025a), GPT-4o (OpenAI, 2024), Gemma-3-12B (Team et al., 2025), and InternVL3-5-8B (Wang et al., 2025b) to evaluate planning and image-based prediction tasks. For text-based evaluation, same proprietary models were used alongside DeepSeek-R1 (Guo et al., 2025), NVIDIA Nemotron-Nano-9B (Basant et al., 2025), and Qwen3-8B (Yang et al., 2025a).

**Metrics.** The scoring process relies on comparisons between ground truth and model-generated text-based outputs. The performance of each task is evaluated using Exact Match, which checks whether all components of the models' answer exactly match the ground truth and computes the percentage of correct predictions. We utilize this metric to assess whether models successfully complete all folding/unfolding steps and generate the correct hole attributes for each problem. To capture cases where models predict the result partially correctly, we include a custom Partial Accuracy metric. Since the results involve multiple holes and their components, we include a penalty for predicting more holes than the correct answer. Given the number of ground truth $G$, the number of the prediction result $P$, and

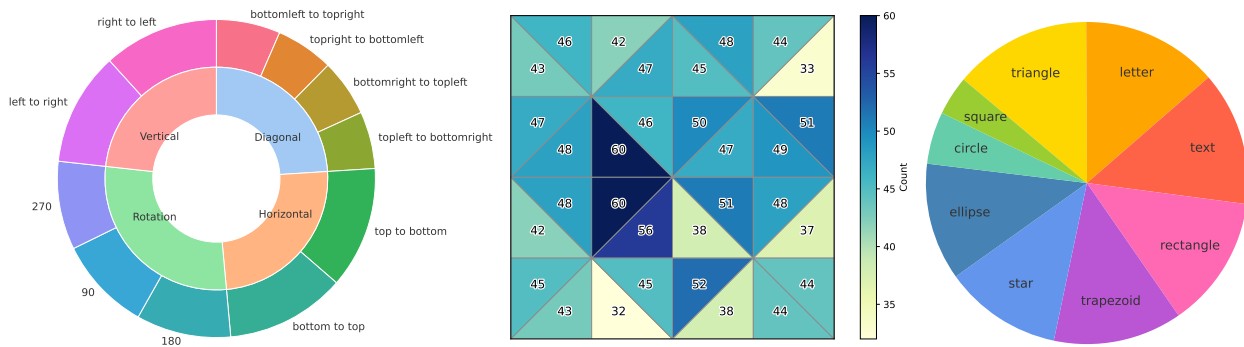

Figure 3: Benchmark statistics of the prediction task. (a) The distribution of the folding and rotation types across four main tasks and eleven sub-tasks. (b) The visualization of the initial hole locations in the paper design. The darker cells represent paper regions with a higher number of punched locations. (c) The distribution of the initial hole shapes across nine categories. Square and circle shapes are excluded from rotation tasks, as their appearance is not affected by rotations.

the number of the matched data $M$, which refers to the count of cases when prediction results exactly match with ground truth data, the final score is calculated as:

$$Partial\,Accuracy = \frac{M}{G + \max(0, P - G)}$$

The scoring method is implemented at two levels: per problem and per hole component. The field-based evaluation measures the correctness of each attribute, size, direction, location, and shape individually, without considering the accuracy of other hole properties in the problem. On the other hand, overall partial accuracy checks all the components of the hole. While field-based partial accuracy provides how many times each attribute is predicted accurately in the problem, overall partial accuracy calculates how many holes are predicted correctly.

Table 2: Model performances across video-based, 2D image-based, and text-based prediction tasks. Metrics are grouped into exact match accuracy, custom score, hole count errors, unfolding accuracy, and hole attribute accuracy. Dashes (–) indicate that the field was not evaluated for that model. Top scores are shown in purple, low scores in red.

| Model | Exact Match | Overall Partial Accuracy | Hole Count Errors (%) | | Unfolding Accuracy (%) | | Field-wise Partial Accuracy (%) | | | |
|---|---|---|---|---|---|---|---|---|---|---|
| | (%) | (%) | Extra | Missing | Exact | Steps | Shape | Size | Location | Direction |
| **Video-Based Prediction** | | | | | | | | | | |
| Qwen2.5-VL | 0.00 | 14.29 | 0.00 | 100 | 3.78 | 16.99 | 22.44 | 23.39 | 17.95 | 21.93 |
| Claude Opus 4.1 | 1.22 | 18.60 | 57.00 | 8.22 | 16.33 | 38.66 | 53.87 | 53.98 | 29.69 | 36.50 |
| Claude 4 | 1.56 | 19.99 | 44.11 | 6.11 | 16.89 | 29.44 | 66.92 | 67.00 | 32.72 | 41.52 |
| LLaVA-OneVision | 0.00 | 19.03 | 0.00 | 100 | 2.78 | 12.69 | 25.49 | 25.95 | 20.32 | 25.41 |
| GPT-4o | 0.33 | 9.97 | 66.56 | 12.78 | 0.78 | 26.02 | 45.65 | 45.96 | 22.23 | 28.66 |
| GPT-5.1 | 1.00 | 18.62 | 34.44 | 12.67 | 18.78 | 37.64 | 67.78 | 68.19 | 30.04 | 44.73 |
| o3 | 8.00 | 28.22 | 32.56 | 10.78 | 24.33 | 44.13 | 67.87 | 68.05 | 41.53 | 47.44 |
| **2D Image-Based Prediction** | | | | | | | | | | |
| GPT-4o | 0.32 | 12.32 | 47.62 | 18.73 | 4.13 | 16.56 | 60.15 | 60.65 | 22.80 | 36.43 |
| Gemma-3-12B | 0.00 | 13.25 | 15.24 | 76.19 | 4.44 | 18.81 | 35.38 | 34.20 | 17.76 | 28.20 |
| LLaVA-OneVision | 0.00 | 18.60 | 0.00 | 100 | 1.59 | 9.40 | 25.38 | 25.65 | 19.14 | 24.78 |
| GPT-5.1 | 0.32 | 18.66 | 30.48 | 22.54 | 5.71 | 22.65 | 70.13 | 70.90 | 28.11 | 45.55 |
| Claude 4 | 0.95 | 19.97 | 41.59 | 17.78 | 8.57 | 27.55 | 62.62 | 62.85 | 29.68 | 40.57 |
| Claude Opus 4.1 | 2.54 | 20.93 | 44.44 | 8.57 | 8.25 | 26.75 | 61.13 | 61.13 | 31.46 | 41.59 |
| o3 | 10.48 | 29.41 | 29.84 | 19.37 | 20.95 | 35.39 | 61.70 | 61.70 | 41.01 | 46.51 |
| **Text-Based Prediction** | | | | | | | | | | |
| Nemotron-Nano-9B | 0.00 | 16.56 | 4.76 | 92.06 | 1.27 | 16.82 | 26.99 | 27.30 | 17.89 | — |
| GPT-4o | 0.32 | 20.42 | 48.89 | 18.10 | 3.81 | 18.41 | 60.25 | 60.29 | 22.81 | — |
| Qwen3-8B | 0.32 | 15.45 | 8.57 | 83.17 | 3.49 | 19.21 | 31.78 | 32.40 | 16.54 | — |
| DeepSeek-R1 | 0.32 | 14.74 | 1.59 | 96.19 | 2.54 | 17.48 | 23.67 | 25.01 | 16.14 | — |
| GPT-5.1 | 3.17 | 25.51 | 39.05 | 26.67 | 6.35 | 22.78 | 62.84 | 63.01 | 28.71 | — |
| Claude 4 | 12.38 | 30.90 | 48.25 | 6.03 | 12.06 | 34.44 | 58.84 | 59.17 | 34.49 | — |
| Claude Opus 4.1 | 17.46 | 33.75 | 46.35 | 7.30 | 15.24 | 30.73 | 60.77 | 60.77 | 38.20 | — |
| o3 | 25.07 | 47.13 | 48.25 | 7.94 | 25.40 | 48.74 | 67.40 | 67.51 | 49.58 | — |

# 5 EVALUATION RESULTS

A preliminary test has been conducted on a couple of proprietary models to analyze whether they understand the content of the visual and textual input used in the prediction and planning tasks. The questions and results are provided in Appendix B. For evaluations prompts, please refer to Appendix G. A human performance test was also conducted to reveal the gap between the models and human participants. The results are provided in the Appendix E.1.

## 5.1 PREDICTION

We evaluate the spatial visualization ability of current models on the prediction task with three formats. Table 2 presents the exact match and partial scores, count errors on result holes, and unfolding accuracy based on both exact and order-sensitive matching. Our evaluation shows that open-source models, which achieve a maximum of 4.4% accuracy on exact fold matching, are not able to reverse the symmetrical folding actions to unfold the papers, despite clear instructions in the text prompt. InternVL3 is not able to complete the task because of the struggle to establish multistep logical reasoning. Also, the high accuracy of open-source models on the missing hole error demonstrates that they struggle to create the spatial objects in the paper space. Partial accuracy on hole properties (below 35%) indicates that capturing spatial attributes is another challenge for open-source models.

Although there is a performance gap between baseline proprietary models and open-source models, the best-performing model, **o3, achieved 25% exact match accuracy.** While they produce fewer errors on missing result holes, they generate a high number of extra holes. Possible reasons are: (1) generating more unfolding steps than the solution; (2) calculating the symmetry without considering physical conditions where the punched area on the top layer does not contain underlying folded layers beneath it, such as a diagonal fold from top right to bottom left, followed by a vertical fold from left to right; or (3) applying the symmetry wrongly by computing more spatial relations based on the crease line. The extra holes reduce the accuracy of the partial score as they are being penalized in the metric computation. The partial accuracies of the proprietary models are above 53% for shape and size, which are easier to predict than others. Location and direction are identified with transformations. The low partial accuracy scores show that symmetry is a challenging task for models. The contrast between component-based and overall scores reveals that the model often captures individual properties but struggles to combine them into a fully coherent prediction.

The performance gap between the exact match unfolding score and the step-based unfolding score, which matches the correct position of the unfolding types in the solution set, implies that **models struggle with sequential reasoning (comprehending the sequence of folds) and visuospatial working memory (following a series of fold actions).** The low exact match accuracy, despite the correct sequence of unfold types, reveals that models fail to implement symmetry transformation. Figure 4 displays the performance loss of o3 models during the unfolding process in nine distinct configurations where the rotation action is included in Groups 5 to 9 (see Appendix C). When the number of folds increases, the difficulty of the problem rises directly. Additionally, a combination of folds and rotations decreases both unfolding and hole prediction accuracies. This result shows that **rotation creates a physical challenge to the models, where they need to understand the orientational and directional alteration of the paper design.** Among the different input tasks, the models perform better on text-based prediction. Although o3 and Claude Opus 4.1 achieve comparable accuracy in exact unfolding for video and text

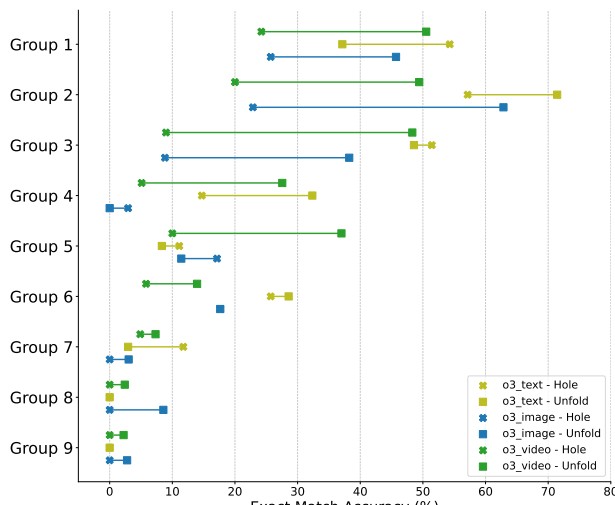

Figure 4: Group-wise comparison of the best-performing o3 models with exact match of holes and unfolding steps.

predictions, they diverge in their exact match scores. Because of the limitation of encoding letters, the text-based structure does not include a direction feature, which reduces the difficulty level of the problem. Although three representations produce similar field-wise accuracy, the overall partial accuracy of the text-based task is around 1.5 times higher than that of the other tasks. The result demonstrates that textual representation mitigates the mental transformation process with a symbolic format.

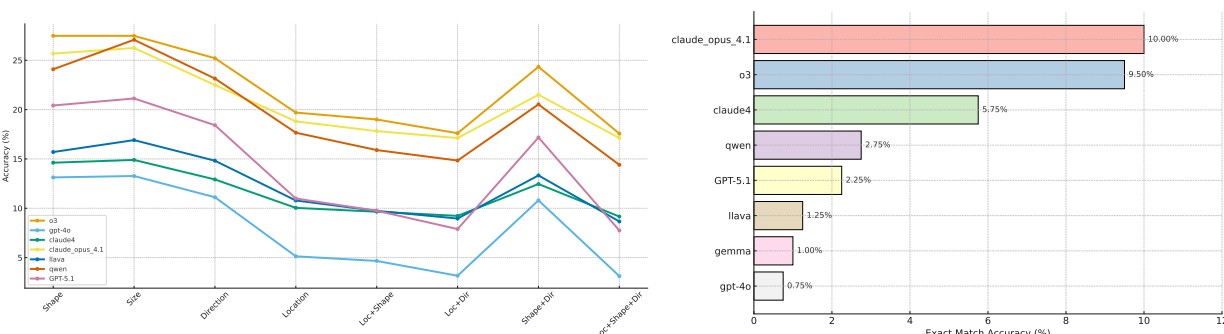

Figure 5: Performance of the models on the planning task: (a) partial accuracy per field; (b) exact match accuracy of baseline models.

## 5.2 PLANNING

For the planning task, a model is asked to provide a series of folds and initial hole properties as the outcome. Since the problem can be solved by using various folding combinations and hole data, we execute the configurations on models' responses on the 3D animation system, which we developed to generate the MantalBlackboard dataset. Each run generates the final hole attributes that belong to the models' planning task. The resulting holes are evaluated to determine whether the holes generated by the model matches the actual outcome. To prevent the simple one-fold planning results, we restrict models to generate the requested count of folds with up to two initial holes. **Claude Opus 4.1 is the best-performing model of the planning task with 10%**, followed by o3, which obtains 9.5% exact match accuracy (see Figure 5). In this task, we observe a high count of missing holes on proprietary models as opposed to the prediction task (see Table 2). Possible reasons are: (1) model preference for fewer steps than the required count in order to obtain a simple solution; (2) a lack of physical awareness of cases where, after multiple steps, the top layer covers the empty area; (3) ordering the physically impossible folds for the paper structure; and (4) failure to mentally simulate folding actions correctly. As a result of the count of hole errors, the partial accuracies of the fields are lower than the prediction task, below 25%. In the planning task, calculating the correct positions of the holes is the primary challenge, followed by determining the direction. The result in Figure 5 reveals that models fail to locate the initial holes, which results in generating a different hole pattern than the solution. Analysis on each task configuration demonstrates that most of the models perform better on one-fold planning problems, except for o3, which achieves better on two-fold planning, the same as the prediction task 2. The complexity level boosts when the number of folding steps increases, which shows that models mentally process a limited number of series of actions.

## 6 DISCUSSION

### 6.1 DO VLMS TRANSFER SPATIAL INFORMATION?

To measure the models' spatial knowledge transfer ability, we create a generalization task, which includes 240 questions in a 2D-image format and three configurations: one-fold, two-folds, and one-fold-one-rotation. The task employs visual analogical reasoning framework with two sequential sets of images. The initial set consists of folding and punching through ordered images, and the result of the unfolded paper. The second set contains identical folding actions but different hole data. The objective is to understand the relationship between the two cases and deduce the spatial components of the resulting holes from the second set. To reach the correct hole pattern, models do not need to unfold the paper and apply knowledge about symmetry; rather, they need to understand the spatial relations among scenarios. The task consists of four categories: size, location, direction, and shape. For each analogy question, only *one type* of hole information is altered at a time. Unlike prediction and planning, the generalization task does not require spatial visualization; it assesses the transfer skills of spatial data. An example of the task is given in Figure 6.

The evaluation results in Figure 7 demonstrate the performance gap between proprietary and open source models on spatial transfer ability. In this task, models tend to underpredict holes rather than overpredict. While o3 accurately solves 71% of the spatial generalization problems, followed by Claude Opus 4.1 and Claude 4 with around 63% and GPT-5.1 with 59.58%, InternVL3, an open-source model, achieves 25% exact match accuracy. Our observations reveal that models implement shape and size arrangements more easily than direction and location. Since the folds between the two cases are identical, models do not need to mentally transform the paper, which explains why the scores are

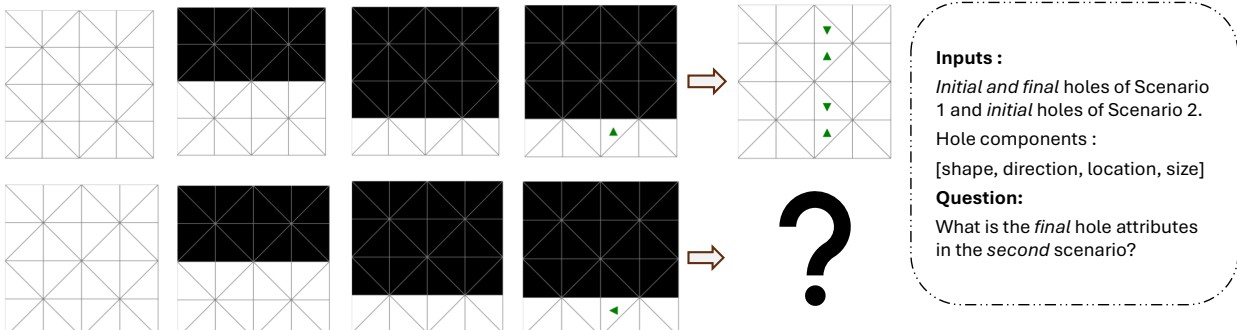

Figure 6: An example of a generalization task employs a visual analogical reasoning framework, which requires making an inference from the first scenario to identify the missing part in the second scenario

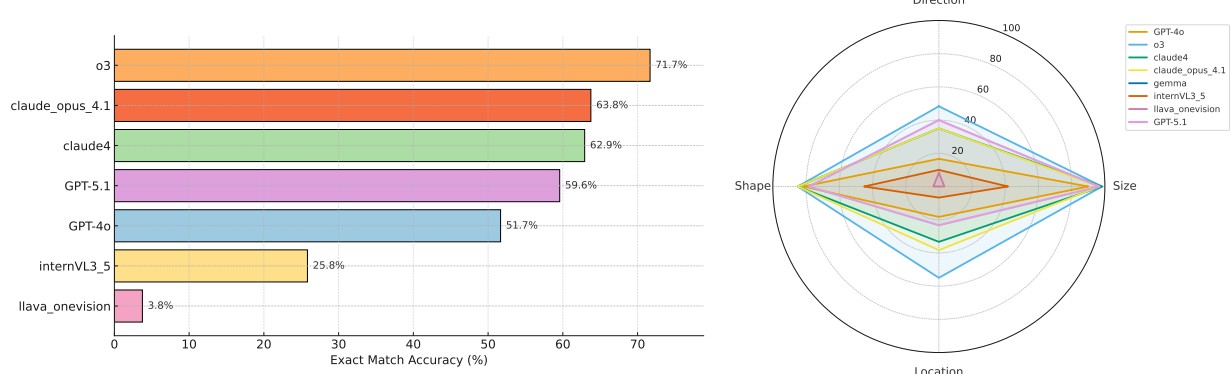

Figure 7: Performance of the models on the generalization task: (a) exact match accuracy of baseline models; (b) the accuracy of the models for each category.

higher than in prediction and planning tasks. However, they still need to calculate the direction and position of the final holes by creating a relation to the result of the first case. Single property change at a time reveals that models mentally rotate the spatial data better in the absence of series symmetry transformations.

## 6.2 HOW DO REPRESENTATIONS INFLUENCE PERFORMANCE?

To discover the impact of textual representation on solving the PFT problem, we re-evaluated model performance on 2D image-based and video-based tasks after removing the direction. Since the 2D image and text-based test sets share identical examples, any difference in performance directly reflects how textual representation influences model behavior in PFT prediction. The video-based dataset, however, includes 900 PFT problems. To include video-based prediction results in the comparison, we extracted the corresponding subset of questions from the text-based cases, resulting in 315 prediction examples per modality.

Five models, o3, Claude 4, Claude Opus 4.1, GPT-5, and GPT-4o, were evaluated across all modalities, and their results were re-analyzed without considering the direction attribute, see Table 3. The findings reveal that textual representations generally yield higher exact-match accuracy than visual formats, except for GPT-4o. Both o3 and Claude Opus 4.1 perform better at the 2D image prediction task compared to video-based evaluations. Notably, Claude Opus 4.1 shows a significant accuracy drop, decreasing from 17.46% on text-based tasks to 4.44% on image-based tasks. Similarly, Claude 4 demonstrates a 10% decrease between text and image-based prediction. But both models show comparable performance between image and video tasks. While GPT-4o shows nearly identical performance across all modalities (0.32%), GPT-5.1 exhibits small variations, with 3.17% accuracy on text and 1.27% on image and video tasks. These results indicate that the Claude models exhibit a substantial performance gap between text-based and visual evaluations.

Table 3: Model Performance across Text, Video, and 2D Image Tasks

| Models | 2D Image | | Text | | Video | |
|---|---|---|---|---|---|---|
| | Exact Match (%) | Overall Score (%) | Exact Match (%) | Overall Score (%) | Exact Match (%) | Overall Score (%) |
| o3 | 19.05 | 39.39 | 25.07 | 47.13 | 15.87 | 40.60 |
| Claude Opus 4.1 | 4.44 | 28.53 | 17.46 | 33.75 | 2.86 | 26.00 |
| Claude 4 | 2.86 | 27.44 | 12.38 | 30.90 | 2.22 | 27.18 |
| GPT-4o | 0.32 | 20.40 | 0.32 | 20.42 | 0.32 | 17.58 |
| GPT-5.1 | 1.27 | 25.62 | 3.17 | 25.51 | 1.27 | 26.37 |

## 6.3 HOW DOES BACKWARD FOLDING AFFECT PERFORMANCE?

The prediction task is performed on forward folding actions, where the folded part of the paper is seen in the video. However, when we change our perspective and do not see the folding actions fully, does it affect the spatial visualization performance? To answer this question, we create a backward prediction task, including 180 video questions with 9 configurations. We implement the same fold types as the prediction task, but the direction of each fold is reversed (away from the camera/observer). To ensure consistency, we adopt the same paper structure, viewpoint, and shadow rendering as forward folding. The camera parameters, including position and zoom, remain unchanged. Consequently, upon completion, both methods yield identical paper visuals and operate within the same action space and spatial representation of holes. The unfolding accuracies shown in Table 17 demonstrate that models o3 and Claude Opus 4.1 visually perceive the backward foldings as similar to the forward foldings in the 2D prediction task. However, they tend to generate more extra holes than in the forward task. Some models, such as Claude Opus 4.1, exhibit a substantial decline in performance when predicting exact unfolding steps in the backward prediction setting, leading to reduced exact match accuracy. This indicates that the model is influenced by view-dependent representations, struggling to Identify the unfoldings when the perspective of the folding action is reversed.

## 7 CONCLUSION

We introduce MentalBlackboard, a large-scale spatial reasoning benchmark designed to assess the spatial visualization ability of current VLMs within prediction and planning tasks. Although many models transfer the spatial data with an accuracy of above 51%, their prediction and planning performance do not exceed 25% and 10%, respectively, which shows that the mental transformation requires a high level of cognitive load. Our evaluations reveal that models struggle to predict the correct sequence of unfolding actions, indicating limited sequential reasoning and visuospatial working memory. Including the mental rotation in the process shows that models lack understanding of the physical alteration in the unfolding method. Although the unfolding steps are predicted accurately, their performance drops during the symmetry process. Also, visualizing backward unfolding introduces another challenge to the mental transformation process for some models. The outcomes of the planning tasks demonstrate that models struggle to perform reverse engineering, which requires understanding the mirror effect on the final hole pattern and imagining the folding actions in sequence. MentalBlackboard reveals the limitations of current VLMs with high-level cognitive tasks. We hope it will contribute to solving the bottlenecks of VLMs for advanced spatial visualization intelligence. Particularly, for R&D groups building next-generation large-scale multimodal and Video GenAI systems, MentalBlackboard provides one more critical perspective: training models to perform complex 2D-to-3D mental transformations, a capability essential for reliable embodied reasoning, robotics, and high-fidelity video generation.

ETHICS STATEMENT

This research poses no ethical issues because it utilizes pre-existing, publicly accessible models without requiring any subjective assessments. The study adheres fully to the ethical standards established by the ICLR Code of Ethics.

REPRODUCABILITY STATEMENT

The benchmark was generated based on the details provided in Appendices C and D. The study's execution can be facilitated by referring to the details in Appendix G. Additionally, we plan to make the data and code freely available under an open-access license, along with detailed instructions that allow the reproducibility of the experiment results, upon acceptance of the paper. The research strictly follows the ICLR Reproducibility Requirements.

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

## A    USING LLMS

In this study, we utilized the LLMs in two ways: code debugging and improving writing at the sentence level. While applying the physical dynamics of the folding and unfolding actions on the 3D animation space in the VPython environment, we encounter some difficulties, such as arranging the lighting of folded paper, which enables better visual perception during the actions. Another difficulty was creating the paper design, which can be folded. When merging the triangle objects as a component of the paper, we encounter implementation issues. To debug the code and similar issues, we utilized LLMs. In paper writing, we employed LLMs for grammar checking.

## B    PRELIMINARY TEST

To test whether the models understand the content of the benchmark data, we created preliminary test cases for each input: single image, series of images, video, and text. For each input data, a varying number of questions are prepared to measure the models' conceptual understanding. While the input of Test 1 consists of multiple punched holes (around 8) and their relations in a single image, Test 3 includes a series of images depicting actions with a couple of marks. In total, four test cases and 37 questions are generated.

**Test 1: Single 2D image understanding** (Planning)
Given the image (see Figure 2), answer the following questions:

1- How many triangles are shown?

2- How many marks with green color are present in the image?

3- What is/are the shape of mark(s) shown in green color in the image?

4- If you count the triangles starting from 1, what is/are the location number of mark(s) in the image?

5- What is/are the angle of the marks(s) in counterclockwise (in degree)?

6- How many unique marks are presented in the image?

**Test 2: Video understanding** (Prediction)
Given the video, answer the following questions:

1- How many sequence of actions are present in the video? (folding, rotation)

2- What are the actions presented in the video in order? (folding, rotation)

3- How many marks with black color are present in the video?

4- What is/are the shape of mark(s) shown in black color in the video?

5- Is ⟨mark⟩ to the left of ⟨mark⟩ in the video?

**Test 3: Series of 2D image understanding** (Prediction and Generalization)
Given the sequence of images, answer the following questions:

1- How many images are given as input?

2- In the first image, how many triangles shown?

3- How many marks with green color are present in the last image?

4- What is/are the shape of mark(s) shown in green color in the last image?

5- Are the marks in the left half of the structure?

6- Are the marks in the right half of the structure?

7- Is ⟨mark⟩ to the left of ⟨mark⟩ in the last image?

8- Is ⟨mark⟩ to the below of ⟨mark⟩ in the last image?

9- If you count the triangles starting from 1, what is/are the location number of green mark(s) in the last image?

10- What is/are the angle of the green marks(s) in counterclockwise (in degree)?

11- In image 2, the count of black triangles are more than the count of white triangles?

12- What is the action taken to go from image 1 to image 2 (folding types, rotations / nothing)

13- From image 1 to image 2, is the object rotated or not?

14- From image 2 to image 3, what is the absolute rotation angle of the object?

**Test 4: Textual content understanding** (Prediction)
Given a text input, the marks are represented by letters. The paper starts as a 4x4 grid of square cells. Each square is divided into two triangles. Each triangle is identified using a location object: [row, column, tri]. Row and col are 0-based indices, starting from the top-left corner of the grid. tri = 0 refers to the first and tri = 1 refers to the second triangle in the square. Punched holes are marked using letters:
'circle': 'C',
'ellipse': 'E',
'star': 'S',
'triangle': 'A',
'trapezoid': 'Z',
'letter': 'T',
Answer the following questions.

1- What is the size of the text array?

2- How many steps shown in the text?

3- What are the letters representing marks?

4- What is the (row, column, tri) location of the mark(s)?

5- In the Step 2, the count of 1 are more than the count of 0?

6- Are the marks in the left half of the grid?

7- Are the marks in the right half of the grid?

8- Is ⟨mark⟩ to the left of ⟨mark⟩ in the array?

9- Is ⟨mark⟩ to the below of ⟨mark⟩ in the array?

10- What is the action taken to go from Step 1 to Step 2 (folding types, rotations / nothing)

11- From Step 1 to Step 2, is the object rotated or not?

12- From Step 1 to Step 2, what is the absolute rotation angle of the object?

**Results.** We evaluate proprietary models, Claude 4, GPT-4o, GPT-5, Gemini 2.5, and o3 on preliminary test cases, whose results are given in Tables 4, 5, 6, and 7. The outcomes of Test 1 display that models fully understand the paper design by counting its components, mostly counting the marks, and recognizing the marks' shapes with minor errors. However, predicting the position and direction of the marks introduces a challenge for models, except for o3. While they correctly calculate these properties for some objects, others can be misplaced. Since they capture the spatial information for a considerable number of marks, the process can be difficult for models.

In the video content understanding Test 2, some models capture the sequence of actions, but have a limited understanding of actions, such as incorrect folding direction or rotation degree. On the other hand, they successfully recognize marks and their shape in video content. The evaluation of a series of images shows that models are able to capture the structure of paper in an abstract representation without fault. The counts and shapes of the marks as well as the relation between them are successfully captured by the models. They recognize the folding and rotating actions between images and provide the accurate rotation angle. However, models struggle to calculate the location and direction of marks. Due to the miscalculations of o3 and GPT-4o, the positioning results are close to 1. The results of the text-based format in Test 4 demonstrate similar success in concept understanding. The locations are expressed differently in the textual representation as row, column, and tri (i.e., which triangle in that location). Although models correctly calculate the row and column, they fail to decide tri, the triangle in which the mark lies, which reduces the positioning of the mark.

To remedy the low performance of locating marks and predicting directions, we include the textual spatial data of each hole mark in the prompts. Although the shapes of the hole marks are mostly recognized by the models, the calculation of the direction and location of the initial marks is not required. Additionally, to make the positioning process easier, we provide the image for visual tasks that displays the numbers for each component of the paper, see Figure 8. For textual location, we describe the value of tri with spatial arrangements, such as left and right in the prompts.

Table 4: Single image results – accuracy (%).

| Test 1 | Claude 4 | GPT-4o | o3 |
|---|---|---|---|
| Q1 | 100 | 100 | 100 |
| Q2 | 42 | 87 | 88 |
| Q3 | 42 | 71 | 83 |
| Q4 | 27 | 50 | 92 |
| Q5 | 13 | 56 | 75 |
| Q6 | 38 | 38 | 50 |

Table 5: Video content results – accuracy (%).

| Test 2 | Claude 4 | GPT-4o | o3 |
|---|---|---|---|
| Q1 | 50 | 25 | 100 |
| Q2 | 25 | 50 | 88 |
| Q3 | 100 | 100 | 100 |
| Q4 | 100 | 100 | 100 |
| Q5 | 100 | 100 | 100 |

## C  DATASET DETAILS

We select three types of paper folding: horizontal, vertical, and diagonal, and three rotation degrees: 90, 180, and 270 to utilize in the dataset. In total, eight fold types are used to create the test problems. As a spatial property, we identify the size, shape, direction, and location. While size consists of two options: small and large, the shapes, selected from the VPython platform, includes nine elements: circle, square, triangle, trapezoid, star, letter, text, ellipse, and rectangle. Directions including degrees of 0, 90, 180, and 270 are used in a counterclockwise way. For the location of the hole, the paper is divided into 32 areas, as illustrated in Figure 8. The dataset configurations are created by combining these folds and rotations. A total of nine distinct structures, five of which include rotation, are generated to create the folding actions; see Figure 8.

Table 6: Series of image content results – accuracy (%).

| Test 3 | Claude 4 | GPT-4o | o3 |
|---|---|---|---|
| Q1 | 100 | 100 | 100 |
| Q2 | 100 | 100 | 100 |
| Q3 | 100 | 100 | 100 |
| Q4 | 100 | 100 | 100 |
| Q5 | 100 | 100 | 100 |
| Q6 | 100 | 100 | 100 |
| Q7 | 100 | 100 | 100 |
| Q8 | 100 | 100 | 100 |
| Q9 | 33 | 16 | 25 |
| Q10 | 42 | 16 | 33 |
| Q11 | 0 | 50 | 100 |
| Q12 | 100 | 100 | 100 |
| Q13 | 100 | 100 | 100 |
| Q14 | 50 | 100 | 100 |

Table 7: Textual content results – accuracy (%).

| Test 4 | Claude 4 | GPT-4o | o3 |
|---|---|---|---|
| Q1 | 100 | 100 | 100 |
| Q2 | 100 | 100 | 100 |
| Q3 | 100 | 100 | 100 |
| Q4 | 50 | 0 | 100 |
| Q5 | 100 | 100 | 100 |
| Q6 | 100 | 100 | 100 |
| Q7 | 100 | 50 | 100 |
| Q8 | 50 | 50 | 50 |
| Q9 | 100 | 100 | 100 |
| Q10 | 50 | 50 | 100 |
| Q11 | 50 | 100 | 100 |
| Q12 | 50 | 50 | 100 |

Figure 8: Paper design with numbered locations.

## D RULES

**Folding rules.** The design of the paper is the key to executing the combination of actions. To utilize the diagonal folds in our dataset, we create a paper structure with 32 triangles. However, some combinations of folds can be infeasible for the current design. Therefore, we create rules that validate the sequence of folding actions regarding the physical structure of the paper. The rules are as follows:

1- No more than two horizontal folds in total.

2- No more than two vertical folds in total.

3- No more than two diagonal folds in a sequence.

4- A diagonal fold is allowed as the first step.

5- A diagonal fold (when not following a diagonal) must come after a combination of one horizontal fold and one vertical fold, or vice versa.

6- A diagonal fold is not allowed, if there is more than one horizontal or vertical fold before it.

7- If there are already two horizontal folds and two vertical folds, then only a restricted set of diagonal types is allowed.

8- Only two of the four diagonal folds are allowed when a second diagonal fold follows the first diagonal fold.

9- The same two diagonal folding directions never come consecutively.

Table 8: Configuration of the MentalBlackboard benchmark. F means a type of paper folding, while R indicates a type of rotation.

| Row | Count of Steps | Rotation | Number of Structure | Action Types |
|---|---|---|---|---|
| Group1 | One | ✗ | 16 | F |
| Group2 | Two | ✗ | 160 | FF |
| Group3 | Three | ✗ | 1408 | FFF |
| Group4 | Four | ✗ | 10752 | FFFF |
| Group5 | Two | ✓ | 48 | FR |
| Group6 | Three | ✓ | 768 | FRF FFR |
| Group7 | Four | ✓ | 10368 | FRFR FFRF FRFF FFFR |
| Group8 | Five | ✓ | 18432 | FRFRF FRFFR FFRFR |
| Group9 | Six | ✓ | 27648 | FRFRFR |

10- After two diagonal folds, exactly one horizontal and one vertical move must follow, or vice versa.

**Folding rules with rotation.** If the folding configuration includes rotation action, the validation rules change accordingly. The sequence of action starts with two steps (one fold and one rotation). The rules on rotation actions are as follows:

1- No rotation is allowed at the first step.

2- Consecutive rotations are not allowed.

3- Diagonal folds are only allowed at the first position.

4- A maximum of three folds in total.

5- If three folds are used, the first must be diagonal.

6- Must have one to three rotations.

**Transformation rules.** Implementation of rotations after folds affects the previous folding actions and changes the crease line during the unfolding process. Instead of re-rotating the paper during unfolding, we require models to understand the physical alteration on the folding line and the orientation of the paper. To predict the unfolding accurately, models should comprehend the impact of the rotation on folding direction and the folding type. The degree of rotation determines whether the folding type or its direction will change during the unfolding process. While 90- and 270-degree rotations change the folding type from one to another, 180-degree rotation leads to the same fold but a different direction. Tables 9, 10, and 11 show the results of how rotation degrees alter folds during the unfolding process.

Table 9: Rotation 90°- counterclockwise

| Folding | Unfolding |
|---|---|
| horizontal top to bottom | vertical right to left |
| horizontal bottom to top | vertical left to right |
| vertical left to right | horizontal top to bottom |
| vertical right to left | horizontal bottom to top |
| diagonal top left to bottom right | diagonal top right to bottom left |
| diagonal top right to bottom left | diagonal bottom right to top left |
| diagonal bottom left to top right | diagonal top left to bottom right |
| diagonal bottom right to top left | diagonal bottom left to top right |

Table 10: Rotation 180°- counterclockwise

| Folding | Unfolding |
|---|---|
| horizontal top to bottom | horizontal top to bottom |
| horizontal bottom to top | horizontal bottom to top |
| vertical left to right | vertical left to right |
| vertical right to left | vertical right to left |
| diagonal top left to bottom right | diagonal top left to bottom right |
| diagonal top right to bottom left | diagonal top right to bottom left |
| diagonal bottom left to top right | diagonal bottom left to top right |
| diagonal bottom right to top left | diagonal bottom right to top left |

Table 11: Rotation 270°- counterclockwise

| Folding | Unfolding |
|---|---|
| horizontal top to bottom | vertical left to right |
| horizontal bottom to top | vertical right to left |
| vertical left to right | horizontal bottom to top |
| vertical right to left | horizontal top to bottom |
| diagonal top left to bottom right | diagonal bottom left to top right |
| diagonal top right to bottom left | diagonal top left to bottom right |
| diagonal bottom left to top right | diagonal bottom right to top left |
| diagonal bottom right to top left | diagonal top right to bottom left |

# E  EXPERIMENT RESULTS

## E.1  HUMAN PERFORMANCE

We evaluated human performance on prediction, planning, and generalization tasks using the Amazon Mechanical Turk platform. We generated sample cases from at most two steps, rotation included PFT problems (Groups 1, 2, and 5). Each human participant answered 4 different questions (video prediction, 2D prediction, text-based prediction, and planning). A total of 64 distinct questions and 16 answers for each category were presented to participants. Human participants were tasked to predict the result of unfolding paper for the prediction task and identify the folding actions for the planning task. The results were collected and merged as JSON files and evaluated against the ground truths. For comparison of human and model performance, we selected two reasoning models, o3 and Claude Opus 4.1. Since human performance is limited in complexity, we retrieved the model results only for groups 1, 2, and 5, and then calculated their average. The table below shows the results. In general, the models achieve higher accuracy for simpler scenarios compared to those that involve several combined folding steps. Across different prediction modalities, human performance remains consistent, 75%; however, the models' performance is highest for textual, then 2D, and lowest for video-based modalities. For the planning task, humans perform similar results to the prediction task, but models struggle to create a sequence of actions with correct hole information. The evident gap in visual prediction between human participants and models indicates that current models have not yet achieved human-level spatial visualization.

Table 12: Model performance across different prediction and planning tasks.

| Model | Video Prediction | Image Prediction | Text Prediction | Planning |
|---|---|---|---|---|
| Human | 75.00 | 75.00 | 75.00 | 75.00 |
| o3 | 16.66 | 21.69 | 40.56 | 14.50 |
| Claude Opus 4.1 | 2.66 | 5.66 | 33.01 | 18.00 |

## E.2  PLANNING

We analyze the performance of the models against each group in the planning task, which does not include rotation. Table 13 shows that the model struggles more when the number of folding steps increases. They can plan one-step

folding tasks better than multiple steps. However, the accuracy of the best-performing model, Cluade Opus 4.1, is 23%. Another analysis on Table 14 shows the percentage of errors for creating holes. Models tend to generate a lower number of holes than expected in the solution.

Table 13: Group-wise performance (% Correct) per model in the planning task.

| Model | One Step (%) | Two Steps (%) | Three Steps (%) | Four Steps (%) |
|---|---|---|---|---|
| o3 | 9 | 20 | 6 | 3 |
| GPT-5.1 | 8 | 0 | 0 | 1 |
| GPT-4o | 1 | 1 | 1 | 0 |
| Claude 4 | 17 | 4 | 2 | 0 |
| Claude Opus 4.1 | 23 | 13 | 4 | 0 |
| LLaVa-OneVision | 3 | 0 | 2 | 0 |
| Gemma | 4 | 0 | 0 | 0 |
| Qwen2.5VL | 9 | 1 | 1 | 0 |

Table 14: Percentage of records with extra and missing hole predictions across models in the planning task.

| Model | Extra Holes (%) | Missing Holes (%) |
|---|---|---|
| o3 | 5.25 | 76.00 |
| GPT-5.1 | 5.25 | 81.00 |
| GPT-4o | 2.75 | 88.25 |
| Claude 4 | 3.25 | 84.00 |
| Claude Opus 4.1 | 5.00 | 73.25 |
| LLaVA OneVision | 7.75 | 81.50 |
| Gemma-3-12B | 10.50 | 64.75 |
| Qwen | 11.75 | 70.75 |

### E.3 GENERALIZATION

The evaluation result of the generalization task in Table 15 demonstrates that most of the open-source models underpredict the spatial hole information and do not transfer the attributes from the first case to the second one. A comprehensive analysis of field-wise accuracy in Table 16 sheds light on possible reasons for the low accuracy scores. Identifying the location and direction introduces more difficulty for models than shape and size arrangements.

Table 15: Extra and missing holes per model in the generalization task.

| Metric | GPT-4o | o3 | claude4 | claude_opus_4.1 | gemma | internVL3_5 | llava_onevision | Qwen 2.5-VL |
|---|---|---|---|---|---|---|---|---|
| Extra Holes | 5 | 5 | 5 | 5 | 30 | 2 | 11 | 9 |
| Missing Holes | 36 | 29 | 30 | 30 | 35 | 119 | 181 | 100 |

Table 16: Hole information accuracy per model (field-wise) on the generalization task.

| Model | Shape (%) | Size (%) | Direction (%) | Location (%) |
|---|---|---|---|---|
| GPT-4o | 81.60 | 84.77 | 78.55 | 67.64 |
| o3 | 83.90 | 87.04 | 83.12 | 82.98 |
| claude4 | 84.01 | 87.06 | 81.85 | 79.44 |
| claude_opus_4.1 | 83.25 | 86.29 | 80.71 | 79.82 |
| internVL3_5 | 60.39 | 59.69 | 53.65 | 52.95 |
| llava_onevision | 45.17 | 51.61 | 50.00 | 48.79 |
| gemma | 16.22 | 32.68 | 29.24 | 5.90 |
| Qwen 2.5-VL | 10.20 | 16.62 | 18.08 | 2.04 |

### E.4 BACKWARD PREDICTION

The backward prediction task evaluates the models' ability to reason on a partially seen sequence of folding actions. Although the structure of the paper does not change, it affects how the action is perceived. The paper is also unfolded from the unseen part, which introduces a challenge to apply the symmetry transformation, as shown in Table 17.

Table 17: Results of the backward prediction task. Step-based unfolding score matches the correct position of the unfolding types in the solution set

| Model | Partial Accuracy (%) | Hole Count Errors (%) | | Unfolding Accuracy (%) | |
|---|---|---|---|---|---|
| | | **Extra** | **Missing** | **Exact** | **Steps** |
| o3 | 26.79 | 41.11 | 13.33 | 20.56 | 34.19 |
| GPT-5.1 | 20.49 | 28.89 | 22.78 | 12.78 | 31.63. |
| Claude Opus 4.1 | 6.51 | 70.56 | 7.22 | 6.67 | 26.28 |
| InternVL3 | 7.13 | 0.56 | 94.97 | 4.47 | 7.73 |
| LLaVA-OneVision | 5.98 | 41.34 | 58.66 | 0.00 | 18.27 |
| Qwen2.5-VL | 10.19 | 0.56 | 99.44 | 6.15 | 17.56 |

### E.5 PREDICTION

The complexity of the prediction task can be categorized according to the number of folding and rotations. When the folding step increases, the requirement of sequential reasoning and visual-spatial memory increases. To evaluate the models' complexity-wise accuracy, we analyze their video prediction results for each folding structure. From groups 1 to 4, the folds consist of diagonal, vertical, and horizontal folds with an increased number of steps. From 5 to 9 rotations are included in the configuration, and start with two steps to six steps. For the group-wise evaluation, we selected o3 and Claude Opus 4.1 as baseline reasoning models since they perform better than other models. Figure 9 shows a detailed analysis of both models against the group structure. When the number of folds increases, the exact match accuracy of the o3 reduces, and the number of missing holes increases. The combination of folds causes o3 to generate extra holes, making a peak at six-fold, including rotation. Although both missing and extra hole numbers decrease close to zero percent in group 5, where folding is followed by rotation, they increase again with the addition of more steps. The precision of unfolding remains consistent, approximately 42%, up to three folds without rotation, but it declines significantly after rotations. These results reveal that rotation increases the complexity level of the tasks along with the number of folds. In contrast to o3, Claude Opus 4.1 initially produces a substantial number of additional holes (approximately 45%) from group 1 to 5, before escalating the amount to 82% during the six-step folding process. In the two-step rotation task, the number of missing holes decreases quickly, whereas extra holes do not show the same reduction. Both the exact unfold and exact match demonstrate a similar decrease as observed in o3; however, unlike o3, the model fails to solve the task that includes the two-step rotation.

To analyze how models perform on one-step fold types (diagonal, vertical, and horizontal) and rotation included cases, we additionally evaluated selected reasoning models, o3, and Claude Opus 4.1 on both video and text-based prediction tasks. The results on Table 2 show that o3 identifies the unfolding actions of horizontal and vertical folds, but struggles with how to unfold diagonal cases. Although the model figures out almost every unfold, excluding diagonal, only half of these scenarios successfully resulted, mostly because of the wrong calculated direction. When we include rotation to these fold types, the unfolding accuracy drops significantly from 90% to 28% for horizontal and 100% to 37% for vertical. As the rotation changes the orientation of the paper, the model should be aware of the physical dynamics of the paper to identify the new unfolding steps. The decrease in unfolding matches reveals that models do not count the orientational alteration of the paper structure. On the other hand, Claude Opus 4.1 performs better on unfolding diagonal folds (45%) but fails on horizontal cases totally, see Table 18. Although o3 correctly predicts shape and size for almost every case, the Claude Opus 4.1 struggles to identify them except in diagonal cases. Unfolding accuracy is reduced by merging the rotation to the folds, 10% for diagonal and 20% for vertical. In general, Claude Opus 4.1 tends to generate extra holes (excluding diagonal folds), unlike o3.

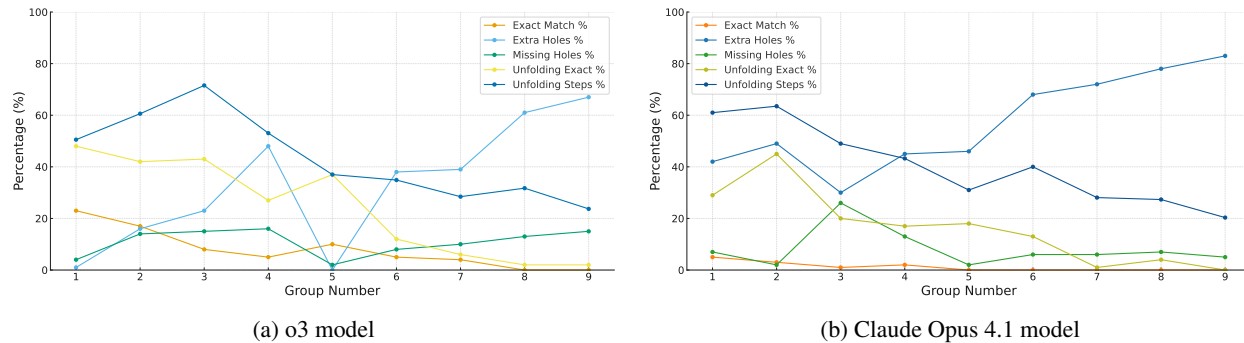

(a) o3 model                (b) Claude Opus 4.1 model

Figure 9: Group-wise evaluation for two models in the video prediction task.

Table 18: Model performances across video-based prediction folds (o3). Metrics are grouped into exact match accuracy, unfolding accuracy, hole count errors, and hole attribute accuracy. D = Diagonal, H = Horizontal, V = Vertical and R = Rotation

| o3 | Exact Match | Unfolding Exact | Extra Hole (%) | Missing Hole (%) | Shape (%) | Size (%) | Location (%) | Direction (%) |
|---|---|---|---|---|---|---|---|---|
| D | 0.00 | 6.25 | 0.00 | 8.33 | 98.6 | 98.6 | 52.1 | 80.0 |
| H | 40.91 | 90.91 | 4.55 | 0.00 | 97.7 | 97.7 | 90.9 | 87.1 |
| V | 56.00 | 100.00 | 0.00 | 0.00 | 100.0 | 100.0 | 100.0 | 89.0 |
| DR | 6.25 | 41.67 | 0.00 | 4.17 | 99.3 | 99.3 | 64.2 | 73.6 |
| HR | 10.71 | 28.57 | 0.00 | 0.00 | 100.0 | 100.0 | 69.6 | 77.4 |
| VR | 16.67 | 37.50 | 0.00 | 0.00 | 100.0 | 100.0 | 66.7 | 83.7 |

Table 19: Model performances across video-based prediction folds (Claude Opus 4.1). Metrics are grouped into exact match accuracy, unfolding accuracy, hole count errors, and hole attribute accuracy. D = Diagonal, H = Horizontal, V = Vertical and R = Rotation

| Claude Opus 4.1 | Exact Match | Unfolding Exact | Extra Hole (%) | Missing Hole (%) | Shape (%) | Size (%) | Location (%) | Direction (%) |
|---|---|---|---|---|---|---|---|---|
| D | 0.00 | 44.90 | 0.00 | 14.29 | 96.6 | 96.6 | 52.0 | 70.1 |
| H | 0.00 | 0.00 | 100.00 | 0.00 | 31.4 | 31.9 | 31.2 | 29.9 |
| V | 20.00 | 28.00 | 64.00 | 0.00 | 57.0 | 57.0 | 48.1 | 52.5 |
| DR | 0.00 | 33.33 | 2.08 | 4.17 | 97.9 | 97.9 | 55.0 | 64.1 |
| HR | 0.00 | 0.00 | 96.43 | 0.00 | 32.6 | 32.6 | 23.6 | 24.6 |
| VR | 0.00 | 8.33 | 75.00 | 0.00 | 47.9 | 47.9 | 34.0 | 36.5 |

## F QUALITATIVE RESULTS

To qualitatively evaluate the model's result, we developed a script that reads the models' JSON outputs and generates a visual representation utilizing the same MentalBlackboard dataset generation environment. This approach allows us to visualize the models' outputs and compare them with the ground truth, unfolded paper results. These visual assessments help us to identify where models fail in generating the answer.

### F.1 PREDICTION

For the prediction task, we examined the cases where models respond correct and incorrect answers. Every example case includes video frames that depict the actions of paper folding and hole punching as the problem description, along with unfolding video frames to demonstrate the solution. Each case also contains the model's example output figure and the ground truth image.

Figure 10 displays an example of the correct prediction task which employs two-step folding actions. The model o3 accurately predicts the unfolding steps as the same as in video frames and applies the symmetry transformation properly to the hole information (size, shape, location, and direction). As a result, the model's visual output matches

the ground truth image. However, Figure 11 shows how the model fails when calculating the directions of the hole during symmetry actions. The example consists of one-step folding with three punched holes. Although the model correctly predicts the unfolding step and calculates the reflection positions of the holes, it fails when calculating the direction of one hole.

Another incorrect case is exemplified in Figure 12 where model o3 predicts the unfolding types incorrectly: Diagonal from bottom right to top left and diagonal from bottom left to top right. The model partially predicts the results and produces missing holes. In other cases, models generate extra holes as results. Figure 13 illustrates the scenario of two-step foldings (diagonal and horizontal), but only one of them creates the reflection of the hole (horizontal). Although the model, o3, identifies the unfolding steps accurately, it applies the symmetry wrong and produces extra holes. In some scenarios, models predict more unfolding steps than the actual condition, which causes extra final holes as in Figure 14. Although the example shows two-step folding actions, Claude Opus 4.1 predicts three unfolding actions (vertical left to right, horizontal top to bottom, and horizontal top to bottom) and creates four extra holes. The physical state of the paper, where the punched area on the top layer does not overlap with any of the underlying folded layers, also leads to extra holes. Figure 15 displays the example where Claude Opus 4.1 predicts the unfolding steps accurately, but it applies the symmetry wrong.

When rotation is combined with folding action, it changes the direction or type of unfolding actions. While the correct scenario of o3 is illustrated in Figure 16, Figure 17 displays the incorrect output of Claude Opus 4.1.

## F.2 PLANNING

To examine how the models generate folding combinations in the planning task, we rendered their outputs as animations. These animations produce both video and image frames representing folding and unfolding actions. The captured frames were then used to visualize the models' outputs for the planning task. Each qualitative example of a planning task contains: the expected final figure, an example of how to solve the task(folding and unfolding video frames), the model's selected folding and unfolding video frames, and the result of the model's reach after running them. If the expected results match with model's output, then the task is answered accurately; otherwise, it is wrong.

Figure 18 illustrates an example of correctly answered planning task that employs two two-step folding. Although the fold types and initial hole information of the model are different than the generated example solution, the expected result matches with model's output. By utilizing the required number of steps and accurate fold-hole integration, o3 reaches the correct result. Figure 19 displays an incorrect example where o3 generates more holes than expected for the task. Since completing the task with a single fold is simpler, some models generate a lower number of steps in contradiction to the requested number of folds. For example, in Figure 20, Qwen2.5VL performs a single fold instead of two steps and uses four initial holes rather than two. Although the expected outcome is not achieved, the model fails even when the results appear to match.

In most cases, models generate missing holes for the planning task. One key reason is that the models fail to compute symmetry correctly when there is an empty space beneath the top layer, see Figure 21. In this example, o3 fails to account for layer depth when combining a sequence of actions, resulting in a partially correct output. Another reason is the model's physically invalid fold combinations. Figure 22 shows how o3 orders folds that do not match with the paper structure in the animation. As a result, the paper becomes distorted and does not place the initial hole on the paper. In some cases, models plan to apply the symmetry wrong with placing the hole not on the folded part of the paper, as in Figure 23. o3 calculates the initial hole location of the trapezoid inaccurately, and the model generates missing holes.

## F.3 BACKWARD PREDICTION

To qualitatively evaluate the backward prediction task, we utilize the same configuration as the prediction task. Figure 24 displays the correct example of the backward prediction task, where o3 predicts the final roles accurately. However, Figure 25 illustrates the incorrect task example where o3 fails to apply the symmetry despite exact match unfolding.

## F.4 GENERALIZATION

In the generalization task, only one attribute changes between two cases, such as location and direction. The model is required to predict the final holes by creating a relationship between cases. Figures 26 and 27 display the correct generalization results. In the first example, direction is the transferred attribute, whereas in the second, it is location. Figures 28 and 29 show that incorrect examples where the model fails to calculate the directions and location accurately.

**Folding :** Vertical right to left, vertical left to right

**Model's Response:**

**Unfolding :** Vertical right to left, Vertical left to right

**Ground Truth:**

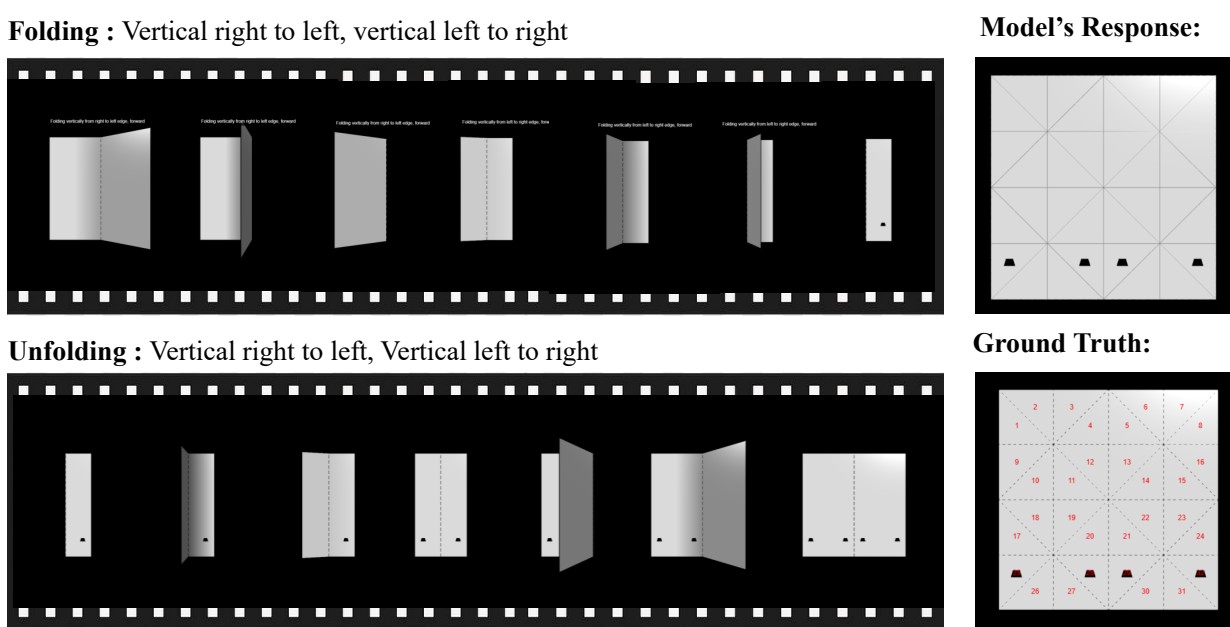

Figure 10: An example of a correct model case for the prediction task

**Folding :** Vertical right to left

**Model's Response:**

**Unfolding :** Vertical left to right

**Ground Truth:**

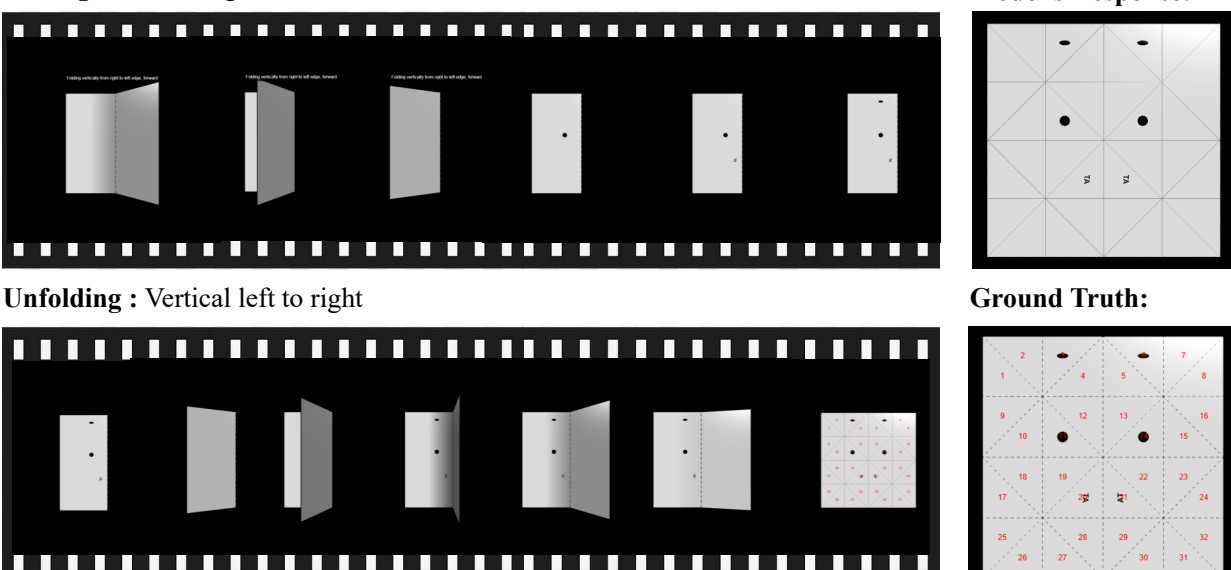

Figure 11: An example of an incorrect case where unfolding actions are accurate but the result is incorrect

**Folding :** Diagonal bottom right to top left, Diagonal bottom right to top left  **Model's Response:**

**Unfolding :** Diagonal top left to bottom right, Diagonal top right to bottom left  **Ground Truth:**

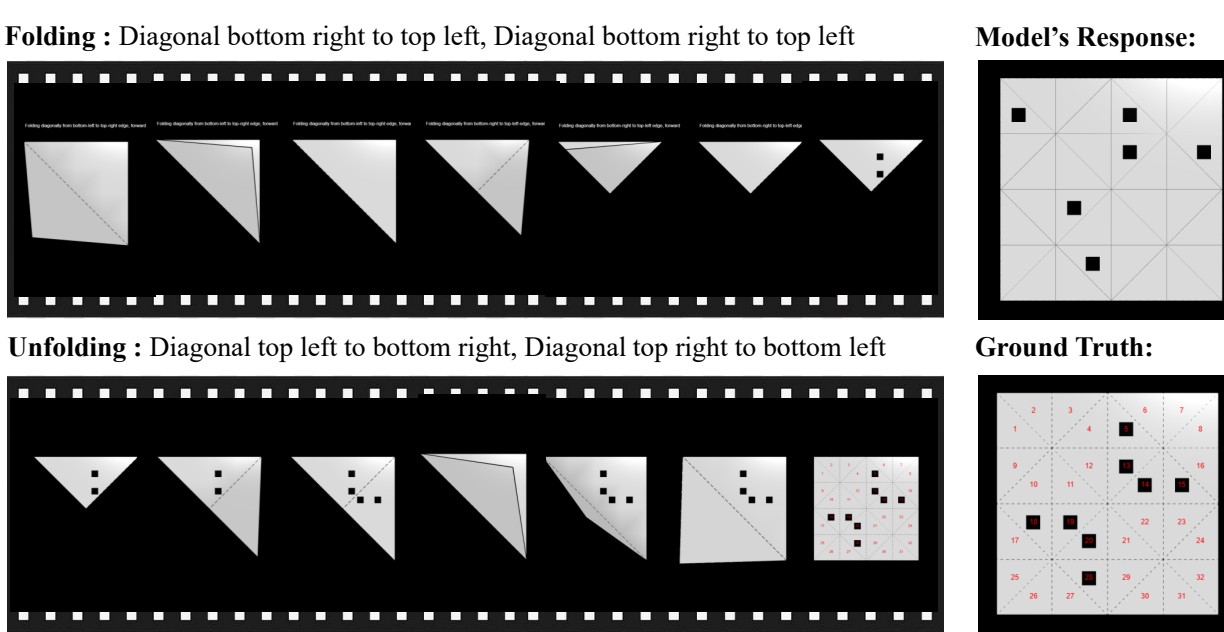

Figure 12: An example of an incorrect case where the model creates missing holes

**Folding :** Diagonal bottom right to top left, Horizontal from bottom to top  **Model's Response:**

**Unfolding :** Horizontal from top to bottom, Diagonal top left to bottom right  **Ground Truth:**

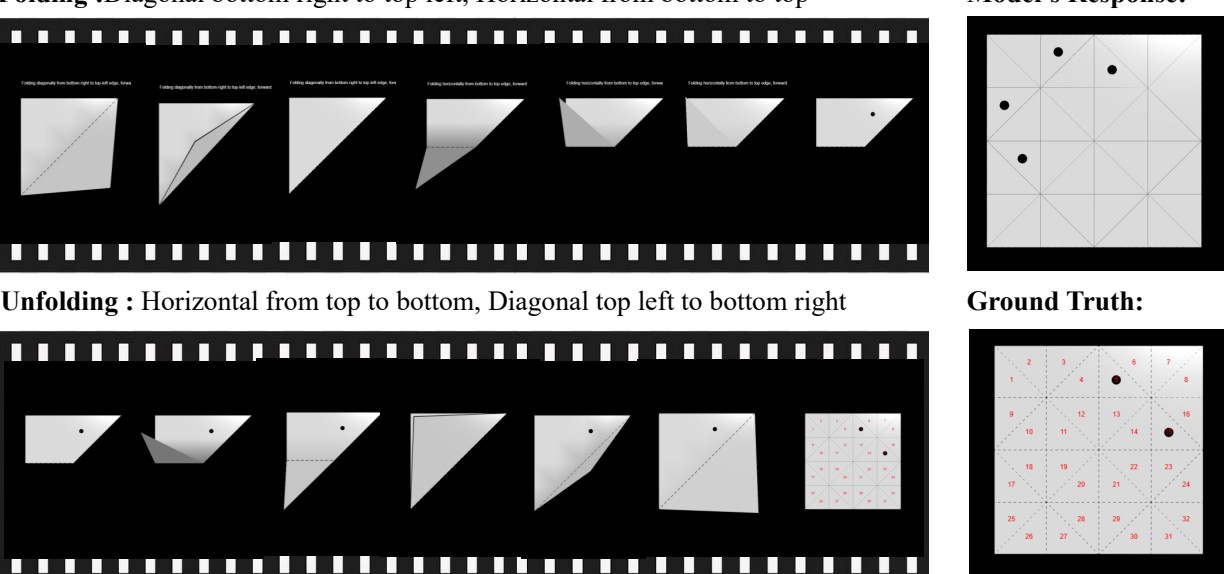

Figure 13: An example of an incorrect case where the model creates extra holes despite accurate unfolding steps

**Folding :** Horizontal top to bottom, Vertical left to right

**Model's Response:**

**Unfolding :** Vertical right to left, Horizontal bottom to top

**Ground Truth:**

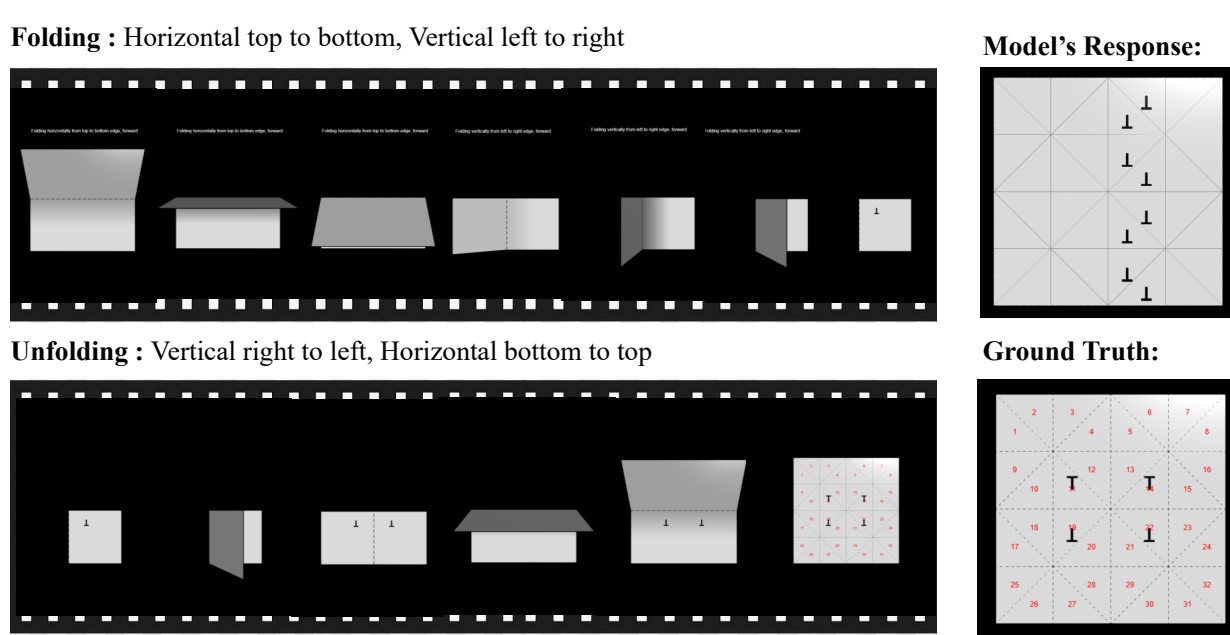

Figure 14: An example of an incorrect case where the model creates extra holes by predicting more unfolding steps

**Folding :** Diagonal top right to bottom left, Horizontal bottom to top

**Model's Response:**

**Unfolding :** Horizontal top to bottom, Diagonal bottom left to top right

**Ground Truth:**

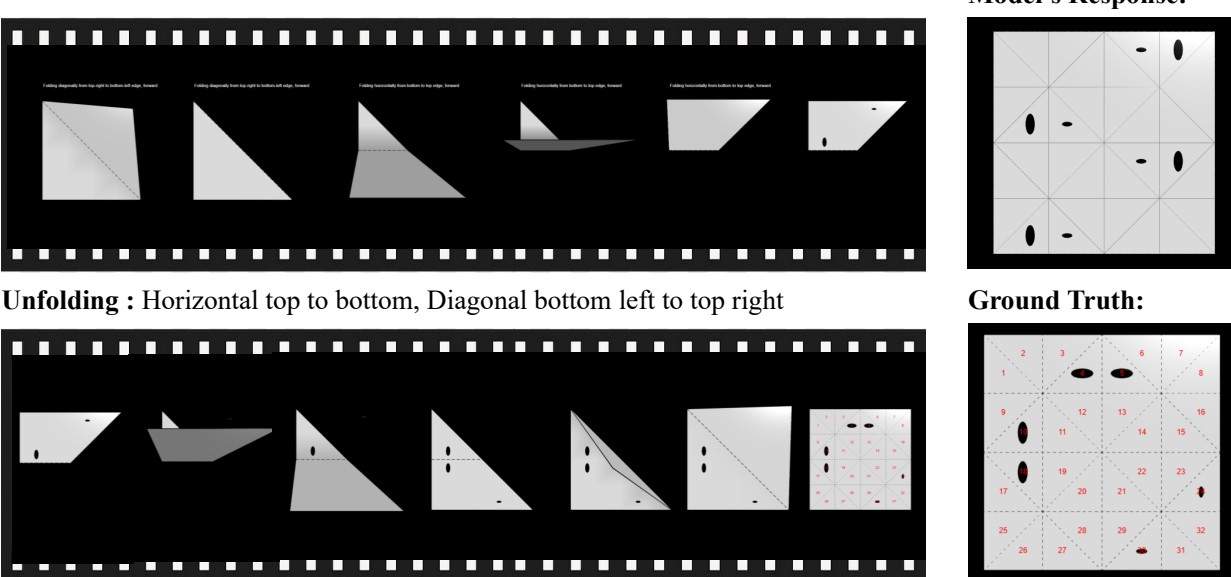

Figure 15: An example of an incorrect case where the model creates extra holes by applying symmetry without considering the physical state of the paper

**Folding :** Diagonal bottom right to top left, Rotation 180

**Model's Response:**

**Unfolding :** Diagonal bottom right to top left

**Ground Truth:**

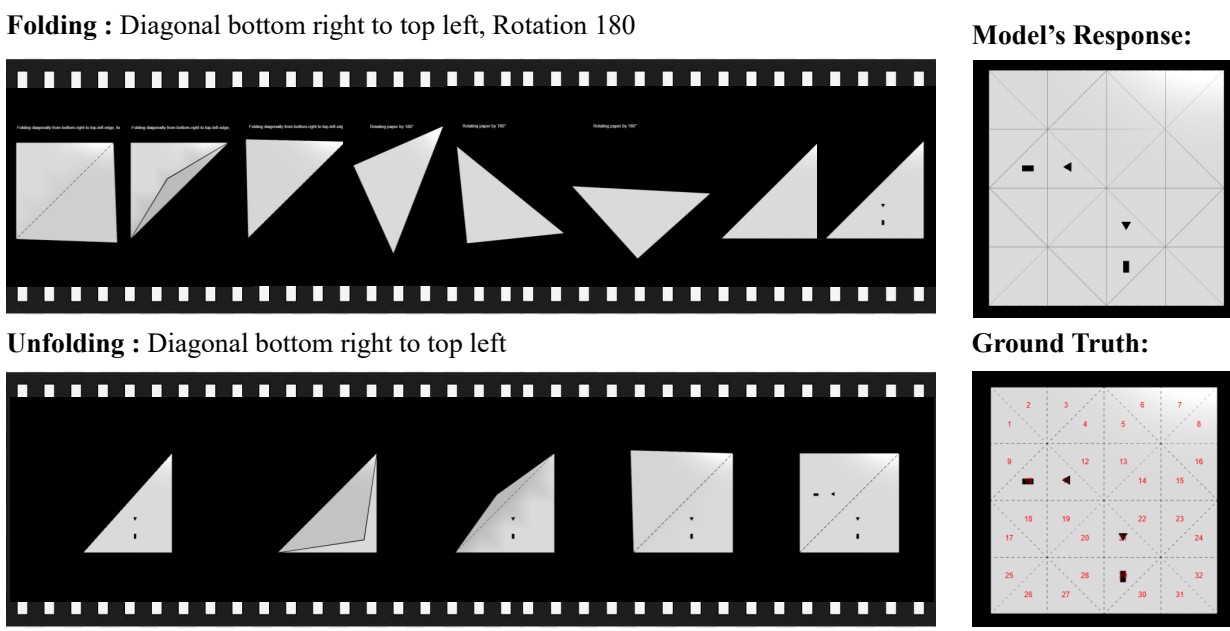

Figure 16: An example of a correct case where the paper is rotated

**Folding :** Diagonal bottom right to top left, Horizontal bottom to top, Rotation 270

**Model's Response:**

**Unfolding :** Vertical right to left, Diagonal top right to bottom left

**Ground Truth:**

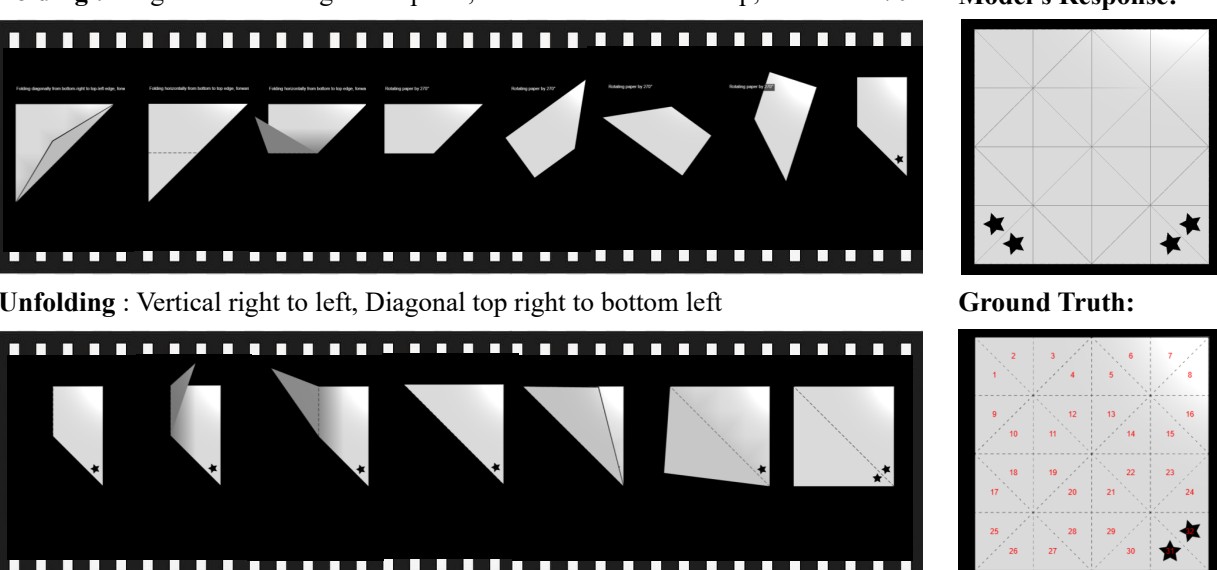

Figure 17: An example of an incorrect case where the paper is rotated

**Expected Result:**

**Example Folding:**

**Example Unfolding:**

**Model's Output:**

**Model's Folding :**

**Model's Unfolding :**

Figure 18: An example of a correct case for the planning task

**Expected Result:**

**Example Folding:**

**Example Unfolding:**

**Model's Result:**

**Model's Folding :**

**o3's Unfolding :**

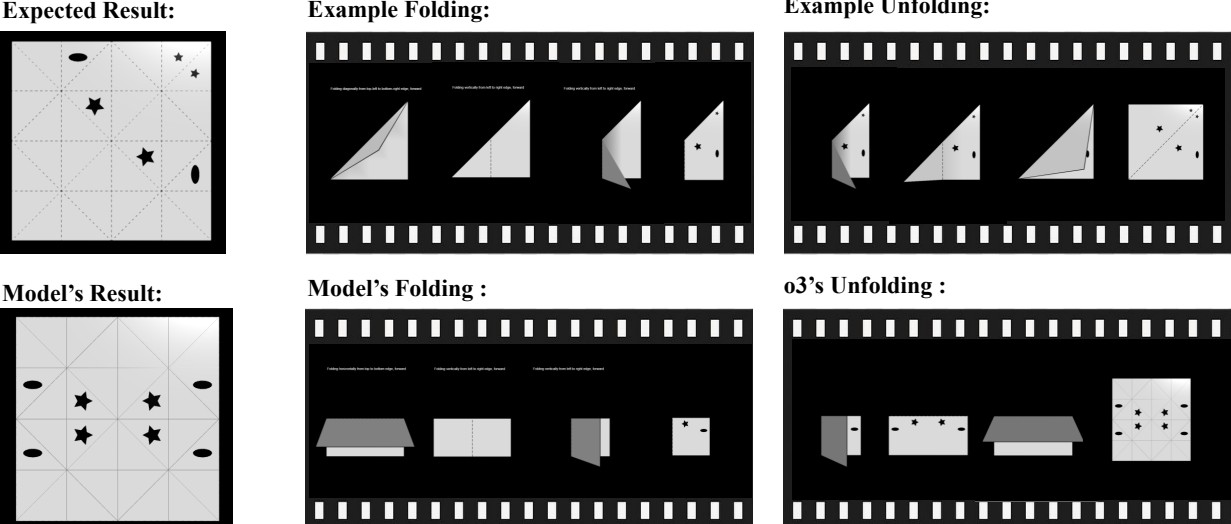

Figure 19: An example of an incorrect case where the model creates extra holes

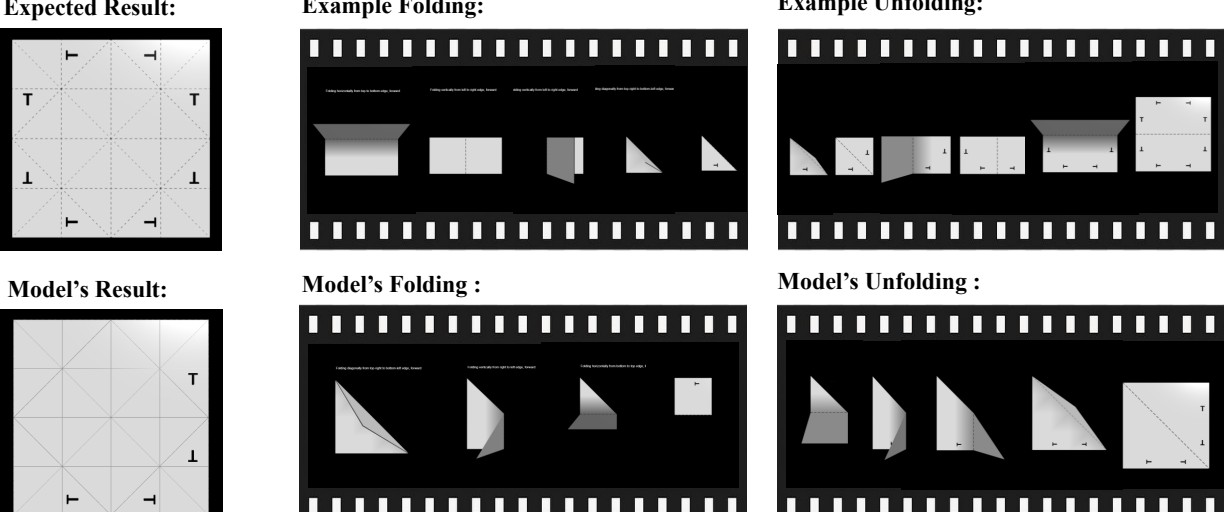

**Expected Result:**

**Example Folding:**

**Example Unfolding:**

**Model's Output:**

**Model's Folding :**

**Model's Unfolding :**

Figure 20: An example of an incorrect case where the model folds paper fewer times than required.

**Expected Result:**

**Example Folding:**

**Example Unfolding:**

**Model's Result:**

**Model's Folding :**

**Model's Unfolding :**

Figure 21: An example of an incorrect case where the physical dynamics of layer depth are not considered.

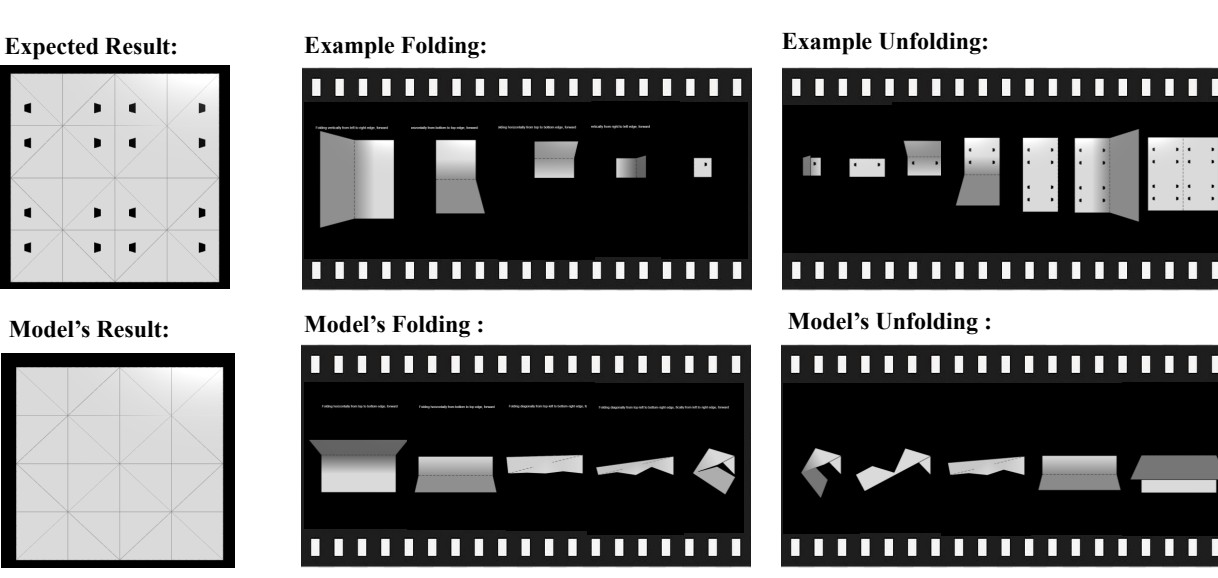

Figure 22: An example of an incorrect case where the model's fold combinations physically distort the paper structure.

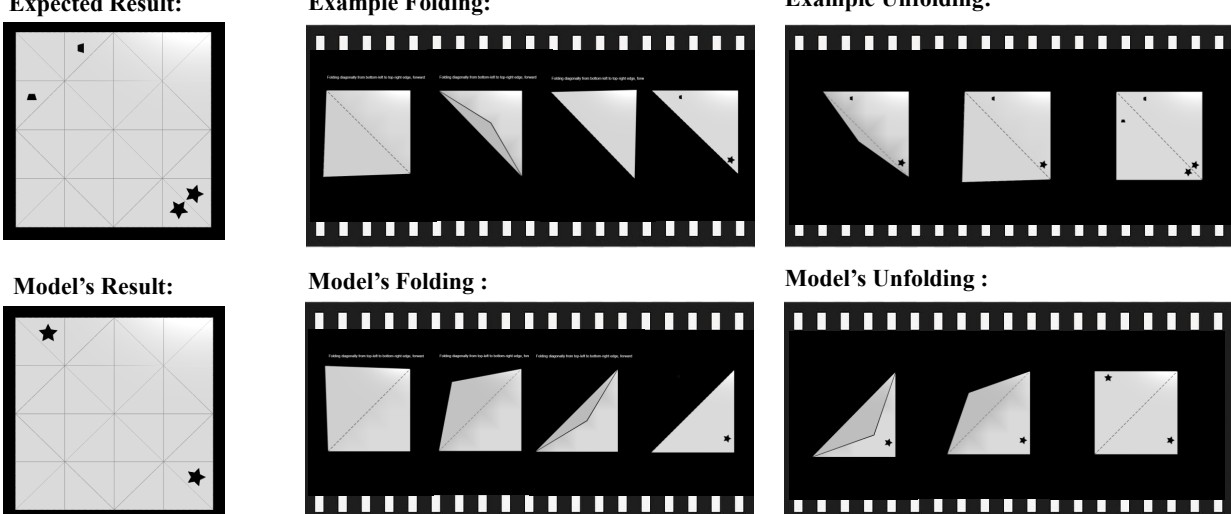

Figure 23: An example of an incorrect case where the model generates missing holes by calculating the initial location of the hole (trapezoid) wrong.

**Folding :** Horizontal bottom to top

**Model's Response:**

**Unfolding** : Horizontal top to bottom

**Ground Truth:**

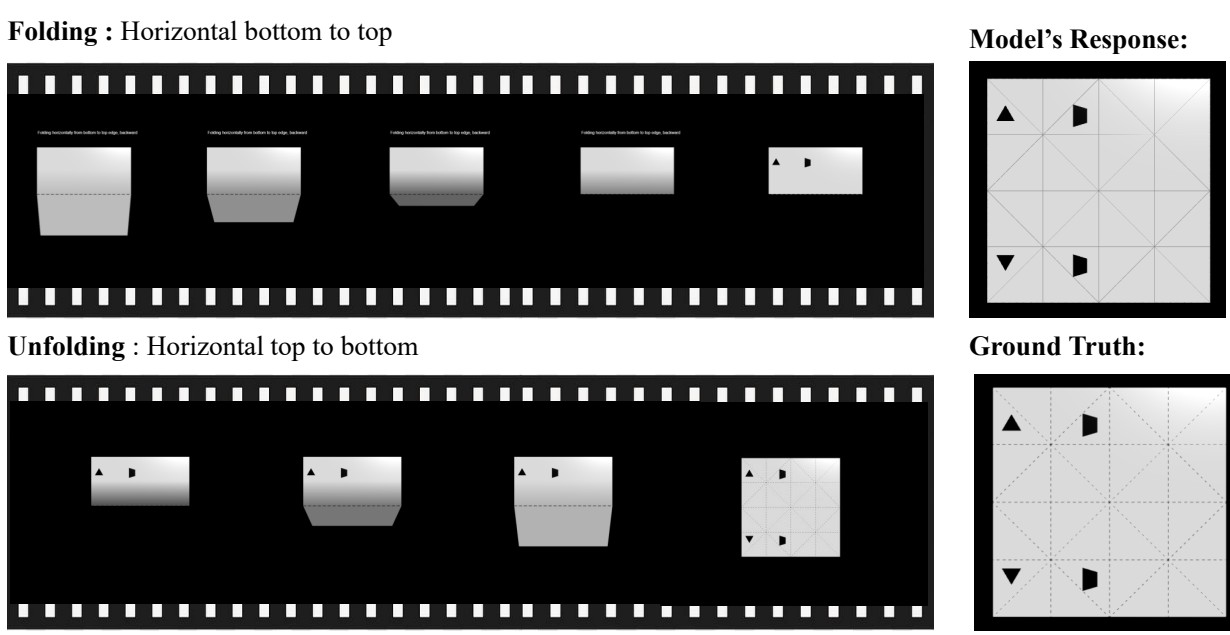

Figure 24: An example of a correct backward prediction task

**Folding :** Diagonal top right to bottom left, Rotation 180

**Model's Response:**

**Unfolding** : Diagonal top right to bottom left

**Ground Truth:**

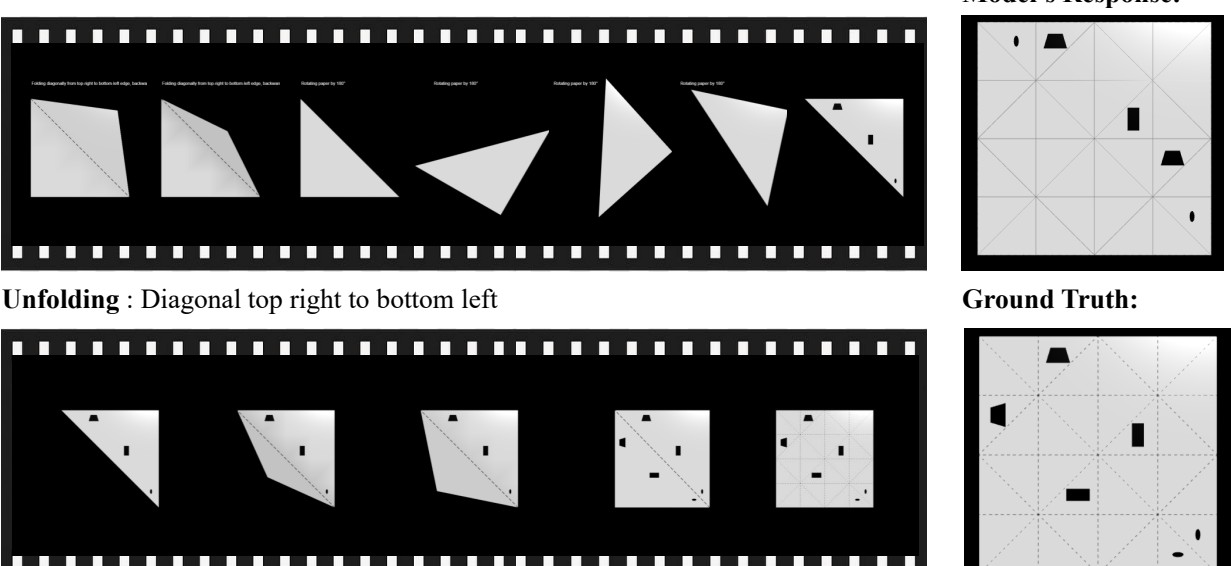

Figure 25: An example of an incorrect backward prediction task

**Generalization:** Direction is different

**Model's Output:**

**Ground Truth:**

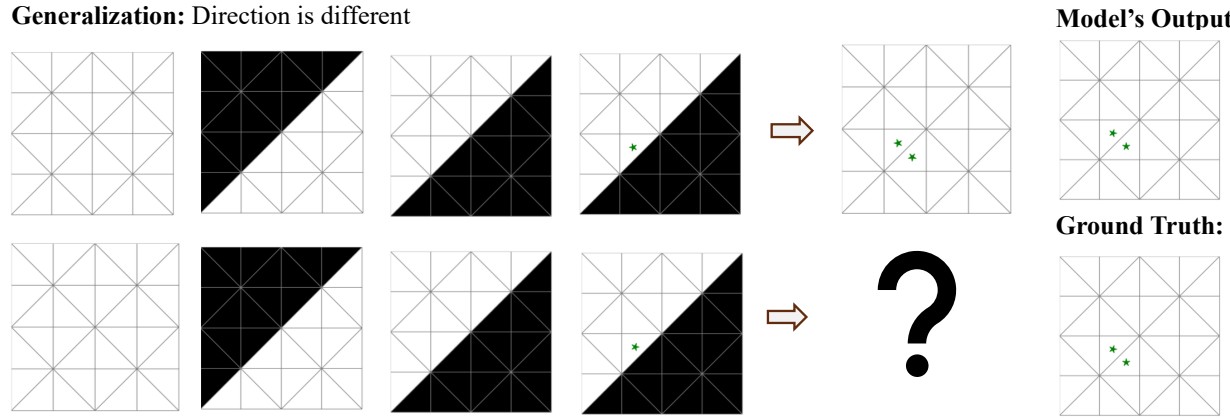

Figure 26: An example of a correct generalization task - direction

**Generalization:** Location is different

**Model's Output:**

**Ground Truth:**

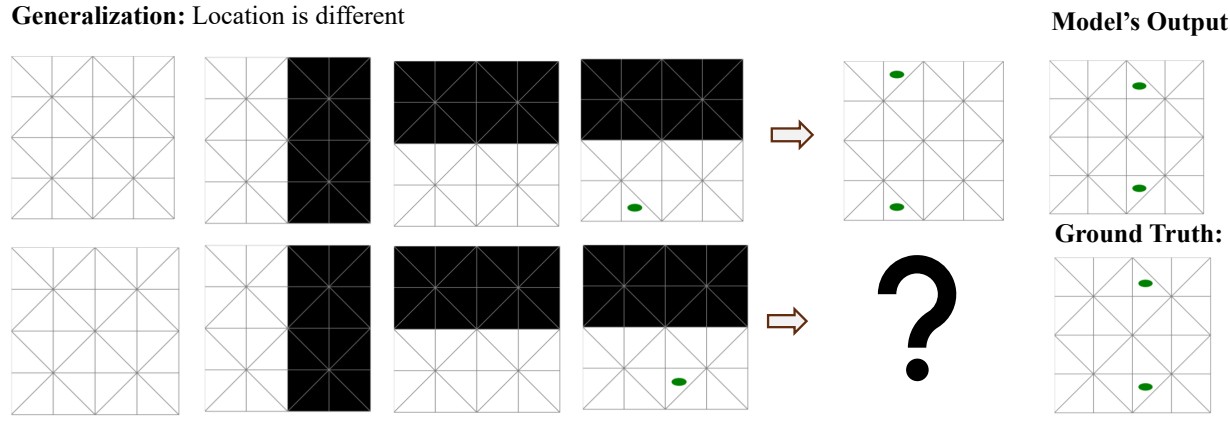

Figure 27: An example of a correct generalization task - location

**Generalization:** Direction is different

**Model's Output:**

**Ground Truth:**

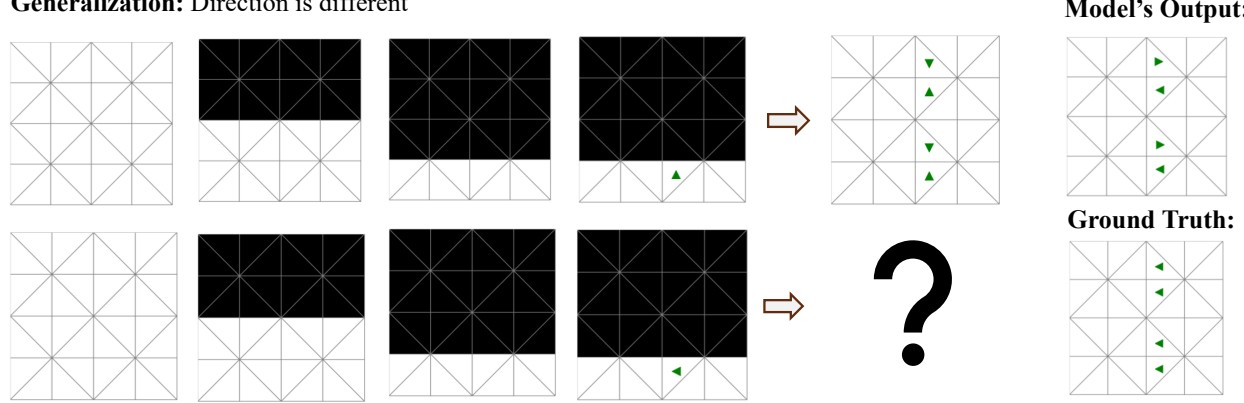

Figure 28: An example of an incorrect generalization task - direction

**Generalization:** Location is different

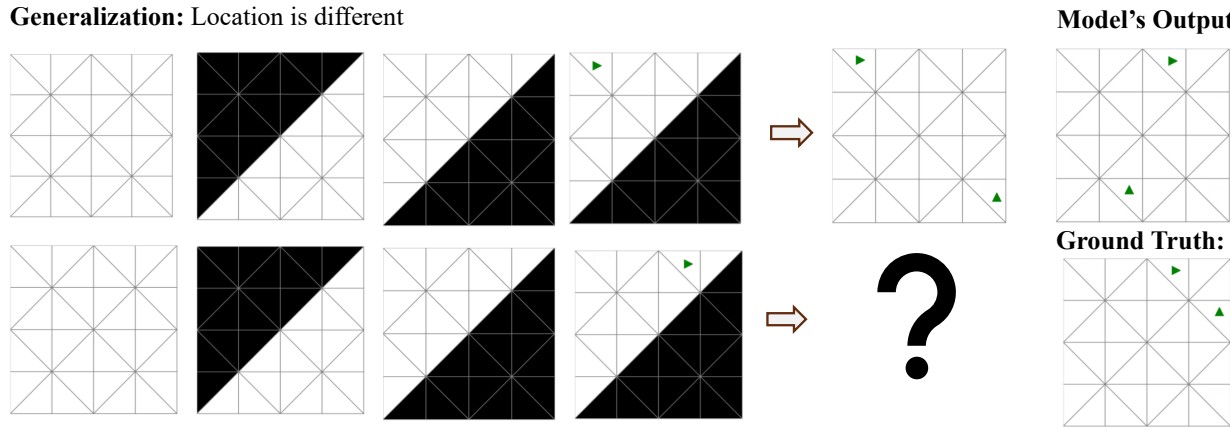

Figure 29: An example of an incorrect generalization task - location

# G PROMPTS

We evaluate the models across two distinct tasks using zero-shot prompting. For each task, a variety of prompts with comprehensive explanations are crafted, along with response options for the requested fields. The details of the prompt template for each task are provided in Figures 30, 31, 32,33, 34, 35, 36, and 37.

---

**Prompt 1: Text-based prediction task**

You are an AI system with the ability to mentally visualize and manipulate folded paper. The following is a textual representation of a paper folding and hole-punching task. The paper starts as a 4x4 grid of square cells. Each square is divided into two triangles. Each triangle is identified using a location object: <row>, <column>, <tri>. Row and col are 0-based indices, starting from the top-left corner of the grid. Triangles are numbered from left to right within each square. tri = 0 refers to the first (left) triangle, and tri = 1 refers to the second (right) triangle.

The paper is folded one or more times according to a defined sequence of images. The paper may also undergo a rotation. Each step in the sequence—excluding the initial state and the hole-punching step—represents either a folding or a rotation action. Rotation affects the physical orientation of the paper and alters the direction of unfolding. Step 0 represents the original unfolded state. In the representation: 0 indicates a folded (hidden) triangle and 1 indicates a visible (unfolded) triangle.

After the final fold, one or more holes are punched in the paper. Punched holes are marked using letters:
    'circle': 'C',
    'ellipse': 'E',
    'star': 'S',
    'triangle': 'A',
    'trapezoid': 'Z',
    'letter': 'T',

The uppercase letters represent large holes and lowercase letters represent small holes. In the given example, the following hole(s) are punched:
Shape: {shape}, Size: {size}, Location: {location}, Direction: {direction} degrees
.

Step 0: initial sheet
11, 11, 11, 11,
11, 11, 11, 11,
11, 11, 11, 11,
11, 11, 11, 11,

Step 1:
00, 00, 00, 01,
00, 00, 01, 11,
00, 01, 11, 11,
01, 11, 11, 11,

Hole Punching:
00, 00, 00, 01,
00, 00, 01, 1T,
00, 0t, 11, 11,
01, 11, 11, A1,

Your task is to mentally unfold the paper step by step. Then, provide: the sequence of unfolding steps, the total number of result holes after unfolding and the final position, size, and shape of each resulting hole on the original paper.

When determining the unfolding steps, do not reverse the rotation. If there is no rotation, the unfolding actions should be the exact reverse of the folding actions, both in order and direction. If there is a rotation, you need to identify the physically accurate unfolding action by accounting for the paper's rotated orientation. Here are your choices for unfolding steps:
  H1-F — Fold horizontally from top to bottom
  H2-F — Fold horizontally from bottom to top
  V1-F — Fold vertically from left to right
  V2-F — Fold vertically from right to left
  D1-F — Fold diagonally from top-left to bottom-right
  D2-F — Fold diagonally from top-right to bottom-left
  D3-F — Fold diagonally from bottom-left to top-right
  D4-F — Fold diagonally from bottom-right to top-left
Think step-by-step, then provide your answer in the required JSON format.
{
 "totalNumberofHoles":<number>,
 "unfoldingTypes": [H1-F | H2-F | V1-F | V2-F | D1-F | D2-F | D3-F | D4-F],
 "resultHoles": [{
     "shape": "<circle | ellipse | triangle | star | letter | trapezoid | square | rectangle| text>",
     "size": "<small | large>",
     "direction": <0 | 90 | 180 | 270>,
     "location": [<row>, <column>, <tri>]
  }
 ]
}

Figure 30: Text-based prediction prompt.

**Prompt 2: 2D image-based prediction task**

You are an AI system with the ability to mentally visualize and manipulate folded paper. Below is a sequence of 2D image representations depicting a paper folding and hole-punching task. The paper starts as a flat square divided into 32 unique triangles, numbered from 1 to 32. They are numbered in order, starting from the top-left corner and increasing row by row from left to right. Last image shows the number locations.

The paper is folded one or more times according to a defined sequence of images. The paper may also undergo a rotation. Each image in the sequence—excluding the initial state and the hole-punching step—represents either a folding or a rotation action. Image 1 shows the original, unfolded paper.

In each image:
Black triangles represent folded (hidden) regions of the paper.
White triangles represent visible (unfolded) regions.
Green shapes indicate holes that were punched after the final fold.

In the given example, the following hole(s) are punched:

Shape: {shape}, Size: {size}, Location: {location}, Direction: {direction} degrees.

Shape: {shape}, Size: {size}, Location: {location}, Direction: {direction} degrees.

.

.

<IMAGE 1>

<IMAGE 2>

.

.

<IMAGE LOCATION>

Your task is mentally unfold the paper step by step. Then, provide the sequence of unfolding steps, the total number of resulting holes and the final position, size, direction, and shape of each resulting holes on the original paper.

When determining the unfolding steps, do not reverse the rotation. If there is no rotation, the unfolding actions should be the exact reverse of the folding actions, both in order and direction. If there is a rotation, you must account for the paper's new orientation before determining the unfolding action. Rotation transforms the axes of the paper, which means the original fold may now appear in a different direction. You need to identify the physically accurate unfolding action based on how the fold is oriented after rotation. Here are your choices for unfolding steps:

  H1-F — Unfold horizontally from top to bottom
  H2-F — Unfold horizontally from bottom to top
  V1-F — Unfold vertically from left to right
  V2-F — Unfold vertically from right to left
  D1-F — Unfold diagonally from top-left to bottom-right
  D2-F — Unfold diagonally from top-right to bottom-left
  D3-F — Unfold diagonally from bottom-left to top-right
  D4-F — Unfold diagonally from bottom-right to top-left

Think step-by-step, carefully mentally unfolding the paper. Then provide your answer in the required JSON format.

```
{
"totalNumberofHoles":<number>,
 "unfoldingTypes": [H1-F | H2-F | V1-F | V2-F | D1-F | D2-F | D3-F | D4-F],
 "resultHoles": [
  {
    "shape": "<circle | ellipse | triangle | star | letter | trapezoid>",
    "size": "<small | large>",
    "direction": <0 | 90 | 180 | 270>,
    "location": <1-32>
  }
 ]
}
```

Figure 31: 2D image-based prediction prompt.

**Prompt 3: Video frame-based prediction task**

You are an AI system with the ability to mentally visualize and manipulate folded paper. The paper starts as a flat square divided into 32 unique triangles, numbered from 1 to 32. These numbers increase row by row from left to right, starting at the top-left corner. The image below shows the number locations:

<IMAGE LOCATION>

The images below are frames from a 3D animation illustrating a paper folding and hole-punching task. The paper is folded one or more times according to a defined sequence of actions. The paper may also undergo a rotation. Each image frame in the sequence—excluding last action which is hole-punching step—represents either a folding or a rotation action.

<VIDEO FRAME 1>

<VIDEO FRAME 2>

.

.

The holes are depicted as black shapes or marks. In the given example, the following initial holes are punched:

Shape: {shape}, Size: {size}, Location: {location}, Direction: {direction} degrees.

Shape: {shape}, Size: {size}, Location: {location}, Direction: {direction} degrees.

.

.

Your task is mentally unfold the paper step by step. Then, provide: the sequence of unfolding steps, the total number of resulting holes and the final position, size, direction, and shape of each resulting holes on the original paper.

When determining the unfolding steps, do not reverse the rotation. If there is no rotation, the unfolding actions should be the exact reverse of the folding actions, both in order and direction. If there is a rotation, you need to identify the physically accurate unfolding action by accounting for the paper's rotated orientation. Here are your choices for unfolding steps:

H1-F — Unfold horizontally from top to bottom
H2-F — Unfold horizontally from bottom to top
V1-F — Unfold vertically from left to right
V2-F — Unfold vertically from right to left
D1-F — Unfold diagonally from top-left to bottom-right
D2-F — Unfold diagonally from top-right to bottom-left
D3-F — Unfold diagonally from bottom-left to top-right
D4-F — Unfold diagonally from bottom-right to top-left

Think step-by-step, then provide your answer in the required JSON format.

```
{
"totalNumberofHoles":<number>,
 "unfoldingTypes": [H1-F | H2-F | V1-F | V2-F | D1-F | D2-F | D3-F | D4-F],
 "resultHoles": [
  {
   "shape": "<circle | ellipse | triangle | star | letter | trapezoid | square | rectangle| text>",
   "size": "<small | large>",
   "direction": <0 | 90 | 180 | 270>,
   "location": <1-32>
  }
 ]
}
```

Figure 32: Video frame-based prediction prompt.

**Prompt 4: Video-based prediction task**

---

<FOLDING VIDEO>

<IMAGE LOCATION>

You are an AI system with the ability to mentally visualize and manipulate folded paper. Above is the 3D animation depicting a paper folding and hole-punching task. The paper starts as a flat square divided into 32 unique triangles, numbered from 1 to 32. They are numbered in order, starting from the top-left corner and increasing row by row from left to right. The image shows the number locations.

The paper is folded one or more times according to a defined sequence of actions. The paper may also undergo a rotation. Each action in the video sequence—excluding last action which is hole-punching step—represents either a folding or a rotation action. The holes are depicted as black shapes/marks.

The holes are depicted as black shapes or marks. In the given example, the following initial holes are punched:

Shape: {shape}, Size: {size}, Location: {location}, Direction: {direction} degrees.

Shape: {shape}, Size: {size}, Location: {location}, Direction: {direction} degrees.

.

.

Your task is mentally unfold the paper step by step. Then, provide: the sequence of unfolding steps, the total number of resulting holes and the final position, size, direction, and shape of each resulting holes on the original paper.

When determining the unfolding steps, do not reverse the rotation. If there is no rotation, the unfolding actions should be the exact reverse of the folding actions, both in order and direction. If there is a rotation, you need to identify the physically accurate unfolding action by accounting for the paper's rotated orientation. Here are your choices for unfolding steps:

  H1-F — Unfold horizontally from top to bottom
  H2-F — Unfold horizontally from bottom to top
  V1-F — Unfold vertically from left to right
  V2-F — Unfold vertically from right to left
  D1-F — Unfold diagonally from top-left to bottom-right
  D2-F — Unfold diagonally from top-right to bottom-left
  D3-F — Unfold diagonally from bottom-left to top-right
  D4-F — Unfold diagonally from bottom-right to top-left

Think step-by-step, then provide your answer in the required JSON format.

```
{
"totalNumberofHoles":<number>,
 "unfoldingTypes": [H1-F | H2-F | V1-F | V2-F | D1-F | D2-F | D3-F | D4-F],
 "resultHoles": [
  {
    "shape": "<circle | ellipse | triangle | star | letter | trapezoid | square | rectangle| text>",
    "size": "<small | large>",
    "direction": <0 | 90 | 180 | 270>,
    "location": <1-32>
  }
 ]
}
```

Figure 33: Video-based prediction prompt.

**Prompt 5: Video-based backward prediction task**

<BACKWARD FOLDING VIDEO>

<IMAGE LOCATION>

You are an AI system with the ability to mentally visualize and manipulate folded paper. Above is the 3D animation depicting a paper folding and hole-punching task. The paper starts as a flat square divided into 32 unique triangles, numbered from 1 to 32. These numbers increase row by row from left to right, starting at the top-left corner. The last image shows the number locations.

 You are always looking at the front side of the paper and all folding actions in the animation are performed by folding the paper backward. The paper may also undergo rotation, which is applied to the entire sheet and is not dependent on front or back orientation. Each action in the sequence—excluding the final hole-punching step—represents either a backward folding or a rotation. The holes are depicted as black shapes or marks.

In the given example, the following initial holes are punched:

Shape: {shape}, Size: {size}, Location: {location}, Direction: {direction} degrees.

Shape: {shape}, Size: {size}, Location: {location}, Direction: {direction} degrees.

.

.

Your task is to mentally unfold the paper step by step, reversing the folds in the correct physical order. Remember that all folds were done backward, so all unfolding actions should also be done by unfolding backward. Provide: the sequence of unfolding steps, the total number of resulting holes and the final position, size, direction, and shape of each resulting holes on the original paper.

When determining the unfolding steps, do not reverse the rotation. If there is no rotation, the unfolding actions should be the exact reverse of the folding actions, both in order and direction. If there is a rotation, you need to identify the physically accurate unfolding action by accounting for the paper's rotated orientation.

Here are your choices for backward unfolding steps:

  H1-B — Unfold horizontally from top to bottom
  H2-B — Unfold horizontally from bottom to top
  V1-B — Unfold vertically from left to right
  V2-B — Unfold vertically from right to left
  D1-B — Unfold diagonally from top-left to bottom-right
  D2-B — Unfold diagonally from top-right to bottom-left
  D3-B — Unfold diagonally from bottom-left to top-right
  D4-B — Unfold diagonally from bottom-right to top-left

Think step-by-step, then provide your answer in the required JSON format.

```
{
"totalNumberofHoles":<number>,
 "unfoldingTypes": [H1-B | H2-B | V1-B | V2-B | D1-B | D2-B | D3-B | D4-B],
 "resultHoles": [
  {
    "shape": "<circle | ellipse | triangle | star | letter | trapezoid | square | rectangle| text>",
    "size": "<small | large>",
    "direction": <0 | 90 | 180 | 270>,
    "location": <1-32>
  }
 ]
}
```

Figure 34: Video-based backward prediction prompt.

**Prompt 5: Video frame-based backward prediction task**

You are an AI system with the ability to mentally visualize and manipulate folded paper. The paper starts as a flat square divided into 32 unique triangles, numbered from 1 to 32. These numbers increase row by row from left to right, starting at the top-left corner. The image below shows the number locations:

<IMAGE LOCATION>

The images below are frames from a 3D animation illustrating a paper folding and hole-punching task. You are always viewing the front side of the paper. All folding actions in the animation are performed by folding the paper backward. The paper may also undergo rotation, which is applied to the entire sheet and is independent of front/back orientation. Each image frame in the sequence—excluding the final hole-punching step—represents either a backward fold or a rotation.

<VIDEO FRAME 1>

<VIDEO FRAME 2>

.

.

The holes are depicted as black shapes or marks. In the given example, the following initial holes are punched:

Shape: {shape}, Size: {size}, Location: {location}, Direction: {direction} degrees.

Shape: {shape}, Size: {size}, Location: {location}, Direction: {direction} degrees.

.

.

Your task is to mentally unfold the paper step by step, reversing the folds in the correct physical order. Remember that all folds were done backward, so all unfolding actions should also be done by unfolding backward. Provide: the sequence of unfolding steps, the total number of resulting holes and the final position, size, direction, and shape of each resulting holes on the original paper.

When determining the unfolding steps, do not reverse the rotation. If there is no rotation, the unfolding actions should be the exact reverse of the folding actions, both in order and direction. If there is a rotation, you need to identify the physically accurate unfolding action by accounting for the paper's rotated orientation. Here are your choices for backward unfolding steps: Here are your choices for backward unfolding steps:

  H1-B — Unfold horizontally from top to bottom
  H2-B — Unfold horizontally from bottom to top
  V1-B — Unfold vertically from left to right
  V2-B — Unfold vertically from right to left
  D1-B — Unfold diagonally from top-left to bottom-right
  D2-B — Unfold diagonally from top-right to bottom-left
  D3-B — Unfold diagonally from bottom-left to top-right
  D4-B — Unfold diagonally from bottom-right to top-left

Think step-by-step, then provide your answer in the required JSON format.

```
{
"totalNumberofHoles":<number>,
 "unfoldingTypes": [H1-B | H2-B | V1-B | V2-B | D1-B | D2-B | D3-B | D4-B],
 "resultHoles": [
  {
    "shape": "<circle | ellipse | triangle | star | letter | trapezoid | square | rectangle| text>",
    "size": "<small | large>",
    "direction": <0 | 90 | 180 | 270>,
    "location": <1-32>
  }
 ]
}
```

Figure 35: Video frame-based backward prediction prompt.

**Prompt 7: 2D image-based planning task**

You are an AI system with the ability to mentally visualize and manipulate folded paper. In this task, you will work with a paper divided into 32 unique triangles, numbered from 1 to 32. Numbering starts at the top-left corner and continues row by row from left to right. To help you identify triangle positions, refer to the reference image below, which shows the numbered layout of the unfolded sheet:

<IMAGE LOCATION>

Below is an image of a paper that has been fully unfolded after undergoing a series of folds and hole punches. In this image, white triangles represent visible (unfolded) areas, and green shapes indicate the final positions of punched holes after unfolding:

<FINAL IMAGE >

In the given example, the following hole(s) appear in the final unfolded result:

Shape: {shape}, Size: {size}, Location: {location}, Direction: {direction} degrees.

Shape: {shape}, Size: {size}, Location: {location}, Direction: {direction} degrees.

.

.

Your task is to determine the sequence of folding actions in the correct order that would produce the given pattern of holes. Then identify the original location, size, direction, and shape of initial hole(s) on the folded paper where the punch(es) were made before unfolding.

You are allowed to punch up to two initial holes on the folded paper. To complete the task, you must use exactly {number} folding steps—no more, no less. Rotation folds are not allowed; only horizontal, vertical, or diagonal folds can be used.

Here are the folding options you may choose from:

    H1-F — Fold horizontally from top to bottom
    H2-F — Fold horizontally from bottom to top
    V1-F — Fold vertically from left to right
    V2-F — Fold vertically from right to left
    D1-F — Fold diagonally from top-left to bottom-right
    D2-F — Fold diagonally from top-right to bottom-left
    D3-F — Fold diagonally from bottom-left to top-right
    D4-F — Fold diagonally from bottom-right to top-left

Think step-by-step, carefully mentally unfolding the paper. Then provide your answer in the required JSON format.

```
{
"totalNumberofHoles":<number>,
 "unfoldingTypes": [H1-F | H2-F | V1-F | V2-F | D1-F | D2-F | D3-F | D4-F],
 "resultHoles": [
  {
    "shape": "<circle | ellipse | triangle | star | letter | trapezoid>",
    "size": "<small | large>",
    "direction": <0 | 90 | 180 | 270>,
    "location": <1-32>
  }
 ]
}
```

Figure 36: Planning prompt.

**Prompt 8: 2D image-based generalization task**

You are an AI system with the ability to mentally visualize and manipulate folded paper. You will be given a visual analogy task involving sequences of paper folding and hole punching operations. Each analogy consists of two parts: a Reference Task and a Target Task. Both tasks will be presented as step-by-step image sequences showing how the paper is folded and where holes are punched.

The paper is represented as a square divided into 32 uniquely numbered triangles, arranged from top-left to bottom-right, row by row. The image below shows the unfolded sheet with numbered triangles to help you locate hole positions:

<IMAGE LOCATION>

In all images:
Black triangles represent folded (hidden) regions of the paper.
White triangles represent visible (unfolded) regions.
Green shapes indicate holes that were punched after the final fold.

Reference Task

This task provides a reference example that demonstrates how a specific sequence of folds and a hole-punching action lead to a particular pattern of holes on the unfolded paper. A series of images will show the step-by-step folding and punching process, followed by a final image displaying the fully unfolded paper with all resulting hole locations.

Folding and punching sequence:
<IMAGE 1>
<IMAGE 2>
.

Unfolded result:

<FINAL IMAGE>

Initial hole(s) after final fold (before unfolding):
Shape: {shape}, Size: {size}, Location: {location}, Direction: {direction} degrees.
Shape: {shape}, Size: {size}, Location: {location}, Direction: {direction} degrees.
.
Resulting hole(s) in final unfolded result:
Shape: {shape}, Size: {size}, Location: {location}, Direction: {direction} degrees.
Shape: {shape}, Size: {size}, Location: {location}, Direction: {direction} degrees.
.

Target Task
This task presents a second sequence of folding and hole punching images, using the same folding sequence as the Reference Task, but without showing the final unfolded result.
Folding and punching sequence:
<IMAGE 1>
<IMAGE 2>
.
Initial hole(s) after final fold (before unfolding):
Shape: {shape}, Size: {size}, Location: {location}, Direction: {direction} degrees.
Shape: {shape}, Size: {size}, Location: {location}, Direction: {direction} degrees.
.

The Target Task uses the same folding sequence as Reference Task. However, only one of the following four aspects of the punched hole(s) is different from the Reference Task: location, shape, size and direction. All other attributes remain the same.

Your goal is by using the visual analogy reasoning, analyze how the folds and punch in the Reference Task produced the unfolded result and then apply the same transformation logic to Target Task considering its different hole information. Determine the final location, size, direction, and shape of each resulting hole in the Target Task on the unfolded paper.

Think step-by-step, then provide your answer in the required JSON format.
```
{
"totalNumberofHoles":<number>,
 "unfoldingTypes": [H1-F | H2-F | V1-F | V2-F | D1-F | D2-F | D3-F | D4-F],
 "resultHoles": [
  {
    "shape": "<circle | ellipse | triangle | star | letter | trapezoid>",
    "size": "<small | large>",
    "direction": <0 | 90 | 180 | 270>,
    "location": <1-32>
  }
 ]
}
```

Figure 37: Generalization prompt.

