# OpenReview forum: "MentalBlackboard : Evaluating Spatial Visualization via Mathematical Transformations"
_ICLR.cc/2026/Conference — ICLR 2026 Conference Desk Rejected Submission_

### Official Review · Reviewer_uNrQ · 2025-10-31

**Soundness:** 3
**Presentation:** 3
**Contribution:** 3
**Rating:** 6
**Confidence:** 4

**Summary:**

The paper proposes a new benchmark called MentalBlackBoard, evaluation framework based on the Paper Folding Test (PFT) to assess the reasoning capabilities of video-language models and vision-language models (VLMs). The authors introduce an automatic pipeline for generating 3D, 2D, and textual representations of the PFT.
The novelty of this work lies in its feasible and large-scale evaluation setup, which encompasses tasks involving memory, relational reasoning, and symmetry understanding. The paper defines two primary tasks.
(1) Prediction, where the model is asked to infer the sequence or attributes of holes after observing the folding steps.
(2) Planning, where the model must generate the correct sequence of folding actions to reach a given final paper configuration.

Experimental results reveal that even SOTA reasoning models (e.g., o3) achieve only about 25% exact-match accuracy, especially when interacting with rich multimodal representations (video or image) for this task. The models fail to reliably understand the symmetry of holes in the folded paper.
Furthermore, performance on the planning task remains very low, with only about 10% success in generating correct folding sequences. The paper also investigates task transferability, showing that while models can recognize hole size and type, they struggle to generalize symmetry and spatial structure in hole patterns.

**Strengths:**

- The paper proposes a benchmark designed to evaluate the ability of vision-language models (VLMs) to understand spatial information using the Paper Folding Test (PFT).

- The experiments are conducted for both forward and backward folding, as well as other action settings, indicating the comprehensive of the evaluation.

- An automatic benchmark creation pipeline is introduced, enabling large-scale dataset generation and ensuring reproducibility.

- The benchmark includes tasks that test both inference and planning abilities, two essential reasoning skills for VLMs. Additional annotations are provided to support detailed performance analysis across different difficulty levels for the inference task.

- The results reveal the limitations of large language and vision-language models in performing this task effectively. Even advanced reasoning models (e.g., o3) achieve only around 10% success on the planning task, highlighting a significant gap in current spatial reasoning capabilities.

- The paper is well-written and easy to follow

**Weaknesses:**

- Figures and their first references are often placed far apart throughout the paper, making it difficult for readers to go back and forth to understand the context.

- The tables are quite large, making it hard to locate specific numbers mentioned in the discussion. Currently, only the overall best results are highlighted across all three settings; it would be clearer to highlight the best results within each setting to improve readability.

- While the authors provide detailed numerical results for analyzing VLM failure cases, there is limited qualitative discussion or example-based analysis to illustrate these failures.

- The paper could be strengthened by incorporating problem-specific prompt engineering or additional strategies to address the current shortcomings observed in both the planning and prediction tasks. Initial approach is fine.

- Although the dataset groups sequences by step for prediction, it would be valuable to include a range of planning problems of varying complexity to evaluate the model’s reasoning progression—from simple to more challenging planning scenarios.

**Questions:**

- It is unclear whether the experimental setup follows a zero-shot or few-shot approach. Additionally, does the VLM have access to information about how to reverse each folding action? Understanding this is important, as it may not be trivial for a model to infer correct unfolding steps directly from a list of candidate actions.

---

> ### Author Response · Authors · 2025-11-21
> **Response to Reviewer uNrQ**
>
> We thank the reviewer for the insightful suggestions on our paper.  We are pleased that you recognize MentalBlackboard reveals the shortcomings of large language and vision-language models with comprehensive and detailed examinations. Below, we provide additional clarifications and qualitative results as requested:
>
> **“Figures and their first references are often placed far apart throughout the paper, making it difficult for readers to go back and forth to understand the context.”**
>
> We reorganized the placement of Figures 1 and 2 in the paper to ensure they appear closer to the sections where they are referred.
>
> **“Highlighting Best Results:”**
>
> Thank you for your feedback. We revised the table accordingly to improve the visual clarity.
>
> **“While the authors provide detailed numerical results for analyzing VLM failure cases, there is limited qualitative discussion or example-based analysis to illustrate these failures.”**
>
> We appreciate the reviewer’s valuable suggestion regarding the inclusion of qualitative analysis. We agree that visual examples of identified error types in both prediction and planning tasks provide a more concrete understanding of the quantitative findings. Although model outputs are based on JSON format, we **developed a script that converts JSON to a figure** in the employed VPython environment. After analyzing the models’ results and finding error cases, we generate the visual representation for prediction tasks. For planning tasks, model-generated action sequences were transformed into our JSON format and animated to retrieve the models’ folding and unfolding videos and outcomes. Since the results are JSON, we converted the JSON to an image to compare whether the model reaches the goal shape or not. We have added several qualitative case studies in Appendix F to better illustrate model behavior. These qualitative scenarios contain correct and incorrect examples from each category: prediction, planning, backward prediction, and generalization. They also cover possible error types stated in sections 5.1 and 5.2 in the main paper. Detailed examples and their explanations can be found in **Appendix F**.
>
> **“The paper could be strengthened by incorporating problem-specific prompt engineering or additional strategies to address the current shortcomings observed in both the planning and prediction tasks. Initial approach is fine.”**
>
> We appreciate the reviewer’s constructive suggestions. In this work, our prompts provided in Appendix G were carefully designed to align with the cognitive structure of the task, requiring the model to mentally visualize and manipulate folded paper while reasoning over sequential actions. This design already integrates domain-specific spatial reasoning elements such as folding directions, rotation handling, and structured JSON outputs. However, we agree that methods like adaptive or task-tuned prompting could further enhance model performance. In future work, we aim to combine prompt-level adjustments with reinforcement learning–based fine-tuning to address the current weaknesses of models in both prediction and planning tasks.
>
> **“Although the dataset groups sequences by step for prediction, it would be valuable to include a range of planning problems of varying complexity to evaluate the model’s reasoning progression—from simple to more challenging planning scenarios.”**
>
> The group-wise performance comparison results of the models in planning tasks are provided in Table 13 in Appendix E.2. For planning, the models require planning the folding actions based on the provided number of steps, which determines the difficulty level. Although each scenario can be completed with a one-step, we increase the complexity level of the task by asking the model to find the exact number of steps to generate the final paper shape. As the task has many possible answers, we restrict models to a number of steps. Since the task does not include rotation action, the number of folds is the only complexity measure. The results in Table 13 show that most of the models perform better at single-step folding planning, except o3, which has a higher accuracy score in two-step folding. The results demonstrate that when the number of steps increases, the models’ accuracy in planning the folding actions drops.
>
> **“It is unclear whether the experimental setup follows a zero-shot or few-shot approach. Additionally, does the VLM have access to information about how to reverse each folding action? Understanding this is important, as it may not be trivial for a model to infer correct unfolding steps directly from a list of candidate actions.”**
>
> The evaluations were performed with a **zero-shot** method. In all prompts, the models are **provided instructions on how to compute unfolding steps based on orientation and rotation.** You can find prompt examples in Appendix G.

---

### Official Review · Reviewer_a6NR · 2025-11-01

**Soundness:** 3
**Presentation:** 2
**Contribution:** 3
**Rating:** 4
**Confidence:** 4

**Summary:**

This paper introduces MentalBlackboard, a new, large-scale, open-ended benchmark designed to evaluate the spatial visualization capabilities of VLMs. The benchmark is based on the Paper Folding Test (PFT) and consists of two primary tasks: prediction and planning .

Unlike previous spatial reasoning benchmarks that often rely on multiple-choice formats , the paper requires open-ended generation, allowing for a deeper analysis of failure modes. The benchmark is procedurally generated using a dynamic 3D VPython pipeline and includes complex transformations like diagonal folds and rotations, which are absent in simpler PFT datasets.

The authors evaluate several proprietary and open-source VLMs. The results are stark: even the best-performing models fail significantly. The top model achieves only 25.07% exact match accuracy on the text-based prediction task and around 10% on the planning task.

Generalization task shows that models can perform good on simple spatial data transfer when the complex, multi-step mental visualization is removed. Backward prediction task shows that performance decreases when folds are partially occluded, highlighting a failure to reason about unseen physical transformations.

**Strengths:**

1. Contribution to Evaluation: This work addresses a clear and important gap. Most spatial benchmarks test static relational understanding or simplified transformations. This paper tackles the much harder, multi-step cognitive process of visualization. The move from multiple-choice to open-ended generation is a major methodological strength, enabling fine-grained error analysis.
2. Insightful ablation: The Generalization task is a particularly clever control; by showing models excel at this visual analogy task (which doesn't require mental unfolding), the authors effectively prove that the core failure is in the multi-step visualization process itself, not just in understanding basic spatial properties.

**Weaknesses:**

1. Speculative Failure Analysis: The analysis of failure modes, while insightful, remains largely speculative. For instance, in Section 5.1, the paper lists several "possible reasons" for why models generate extra holes, but these hypotheses are not empirically validated. The paper would be much stronger if it included analysis to confirm these claims, perhaps by correlating specific error types with specific task configurations.

2. Missing Highly Relevant Related Work: The related work section, while good, overlooks some very recent and highly relevant work. For example,*"Unfolding Spatial Cognition: Evaluating Multimodal Models on Visual Simulations."* and *"SpatialViz-Bench: An MLLM Benchmark for Spatial Visualization"*


3. Limited Argument for Novelty: While the paper's open-ended format and inclusion of rotations are clear improvements, the core task is still the PFT , which is well-established in both cognitive science and recent AI benchmarks (e.g., Mind the Gap, OmniSpatial, SpatialViz, STARE). The paper could do a better job of arguing why its specific additions (like 3D generation and rotation) constitute a qualitatively new cognitive challenge for VLMs, rather than just a more difficult, incremental version of the same underlying problem.


4. Unclear Presentation of Methods and Data: There are a few points of unclear presentation.
 - Action Space: The 'action space' for unfolding (e.g., 'H1-F', 'D2-F') is defined in the Appendix prompts (e.g., ) but should be made explicit in the main paper's methodology section for clarity.
 - Dataset Size: There is a confusing discrepancy in the reported dataset size. Table 1 claims a size of more than 1M , but Section 3 states the pipeline generates "over 12K unique configurations". The authors should clarify if the "1M" refers to total generated frames versus the number of problems.

**Questions:**

1. Model Selection:  Given that GPT-4o is a premier, publicly available vision-language model, why was it only evaluated on 2D image and text inputs, and not on the video-based prediction task? Also performance of gpt-5 should also be included.
2. Need for Qualitative Analysis: The quantitative analysis is thorough, but the paper would significantly benefit from qualitative case studies. Could the authors include examples showing a model's actual incorrect output (e.g., the JSON it produced) versus the ground truth? Seeing a concrete example of a model misapplying a symmetry rule would make the quantitative findings in Table 2 much more concrete. Also error types should be included in case studies.
3. Visual vs. Text Performance Gap: The performance gap between text and vision is striking. The paper notes the text task is simpler (it omits the 'direction' feature). How much of this performance gap do you attribute to that simplification versus a more fundamental failure in visual perception (i.e., models struggling to parse the video/images into a stable symbolic representation)?
4. Human Baseline: To properly calibrate the difficulty of this benchmark, what is human performance on these tasks? Establishing a human-level baseline (even on a small subset of the test data) would be invaluable for contextualizing the models' very low scores.

---

> ### Author Response · Authors · 2025-11-21
> **Response to Reviewer a6NR (Part 1)**
>
> We thank the reviewer for the insightful comments on our paper. We are pleased to see that our work, Mentalblackboard, is recognized for its methodological strength in conducting fine-grained error analysis of the multi-step cognitive processes involved in spatial visualization. Please find our detailed responses and additional experiment results to your specific inquiries below:
>
> **Speculative Failure Analysis:**
>
> We appreciate the reviewer’s observation regarding our initial failure analysis. To strengthen the empirical basis of our claims, we have conducted a quantitative evaluation of the error types. The provided reasons for models generating extra holes are: (1) more unfolding steps than actual, (2) calculating the symmetry wrong, not considering the physical conditions of fold layers, and (3) misapplied symmetry. Besides extra holes, we also investigated the same cases for missing holes. As a baseline model, we selected the two best-performing reasoning models in the video prediction task, o3 and Claude Opus 4.1. The table below shows the evaluation results. Also, Appendix F illustrates the visual examples for each error type model generated.
>
> For case 1, while o3 does not generate more unfolding steps for the video prediction task, **Claude Opus 4.1 predicts more unfolding steps, which account for 19.11% of the error** in the task, though these rarely cause missing holes. Case 2 occurs with diagonal folds followed by horizontal or vertical folds, creating uneven layers that complicate symmetry calculations. The results indicate that both o3 and Claude Opus 4.1 produced extra holes with error rates of 12.78% and 16.89%, respectively, revealing that the **models do not adequately account for the physical layer depth when applying symmetry.** Also, these models do not compute the hole attributes accurately, even though they generate an equal number of hole results (4.33% and 5.33%). Case 3 occurs when unfolding steps are correct but symmetry is applied incorrectly. Both models fail in approximately 5.11% of cases, while missing holes occurring alongside exact unfolding are relatively rare (around 0.62%). This suggests that **although the models can reproduce the geometric unfolding accurately, they still struggle to calculate hole information accurately.**
>
> | Model           |  More Unfolds & Extra Holes  | Fold Depth & Extra Holes  | Exact Unfolds & Extra Holes  | More Unfolds & Missing Holes | Fold Depth & Missing Holes  | Exact Unfolds & Missing Holes  | Fold Depth & Equal Holes  |
> |-----------------------|-------------------------------------------|-------------------------------------------|------------------------------------------------|---------------------------------------------------|-----------------------------------------------|-------------------------------------------------|-------------------------------------------------|
> | o3              | 0.25                             | 12.78                         | 5.32                             | 0                                  | 3.89                           | 0.62                               | 4.33                          |
> | Claude Opus 4.1 | 19.11                            | 16.89                         | 5.11                             | 0.67                               | 2.56                           | 0.67                               | 5.33                          |
>
>
> **Need for Qualitative Analysis:**
>
> We appreciate the reviewer’s valuable suggestion regarding the inclusion of qualitative analysis. We agree that visual examples of identified error types in both prediction and planning tasks provide a more concrete understanding of the quantitative findings. Although model outputs are based on JSON format, we developed a script that converts JSON to a figure in the employed VPython environment. After analyzing the models’ results and finding error cases, we generate the visual representation for prediction tasks. For planning tasks, model-generated action sequences were transformed into our JSON format and animated to retrieve the models’ folding and unfolding videos and outcomes. Since the results are JSON, we converted the JSON to an image to compare whether the model reaches the goal shape or not. We have added several qualitative case studies in Appendix F to better illustrate model behavior. These qualitative scenarios contain correct and incorrect examples from each category: prediction, planning, backward prediction, and generalization. They also cover possible error types stated in sections 5.1 and 5.2 in the main paper. Detailed examples are provided in **Appendix F.**
>
> **Unclear Presentation of Action Space**:
>
> We thank the reviewer for pointing out this issue. The unfolding actions are the same as the folding types provided in Figure 3. We have added a clarification in the Benchmark Construction section.

---

> > ### Author Response · Authors · 2025-11-21
> > **Response to Reviewer a6NR (Part 2)**
> >
> > **Missing Highly Relevant Related Work:**
> >
> > The first version of SpatialViz-Bench [4] was submitted to Arxiv slightly more than two months ago, before the ICLR deadline, and their last version of Arxiv was submitted on September 2. Since the paper is recently publicly available and not published in venues yet, the benchmark did not get our attention in the related work section. After analysis, we found that SpatialViz-Bench reports similar results in Table 1. Its tasks, including Paper Folding, **not combining symmetry, memory, and rotation, remain 2D, use multiple-choice evaluation, and include around 100 samples per task**, similar to other spatial visualization benchmarks. A similar limitation in evaluation structure and perspective is observed in SpatialViz-Bench. On the other hand, MentalBlackboard provides large 3D tasks including multiple mathematical transformations with physical dynamics and fine-grained error analysis.
> >
> > Unlike these datasets, the Unfolding Spatial Cognition [5] paper focuses on spatial reasoning and transformations rather than spatial visualization. Although the dataset does not mention spatial visualization, one task, **Cube Folding,** is an example of this category. The task consists of **320 samples with the mental rotation process represented in 2D dimensions.** The model evaluates a sequence of folding steps to decide whether the net can form a cube, giving a **binary yes/no response.** While this binary evaluation offers finer error tracking than multiple-choice methods, it **limits models to fixed responses and lacks deeper reasoning analysis.** In contrast, reasoning-based generation provides greater interpretability and clearer insight into model weaknesses.
> >
> > The related work section and Table 1 were updated according to these datasets.
> >
> > [4] https://arxiv.org/abs/2507.07610
> >
> > [5]  https://doi.org/10.48550/arXiv.2506.04633
> >
> > **Limited Argument for Novelty**
> >
> > The novelty of our study lies in the development of an automated dataset creation pipeline, which makes the benchmark both scalable and physically consistent. Unlike prior 2D benchmarks, our **rule-based pipeline** (Appendix D) generates data grounded in **physical constraints**, supporting over **12K unique configurations** across eight folding types (Figure 3). We also introduce a fully **3D simulation environment, bridging the gap between synthetic datasets and real-world applications** by modelling spatial and physical features of objects and actions. This makes our benchmark better reflect complex spatial and physical interactions found in real-world tasks.
> >
> > **The revised PFT tasks incorporate rotation transformations, adding physical awareness and complexity beyond traditional reflection-based folds.** To unfold correctly, models must infer crease orientation and transformation effects. Tables 9–11 show how rotation alters unfolding dynamics. In addition to prediction, we developed a **planning task that requires reasoning through sequential transformations, turning a static problem into a dynamic spatial reasoning challenge** where success depends on recognizing outcomes and generating intermediate transformations. This extended PFT framework combines multiple transformations under physical constraints, making it a distinctive and comprehensive benchmark for spatial visualization.
> >
> > **Model Selection:**
> >
> > Thank you for pointing this out. Since OpenAI and Anthropic models lack native video input support, we provided videos as frame sequences and selected their reasoning models for evaluation. Upon request, GPT-4o and Claude 4 were also tested on video prediction using the same frame-based setup, and the newly released GPT-5.1 was included for all tasks.
> >
> > GPT-4o achieved the same exact-match accuracy (0.33%) on video and 2D image tasks, while Claude 4 performed slightly better (1.56%). GPT-5.1 matched GPT-4o on 2D prediction but performed marginally better on video (1%) and text (3.17%) tasks. For planning tasks, GPT-5.1 achieves 2.55% accuracy, similar to the Qwen2.5VL model. In the generalization task, GPT-5.1 outperforms GPT-4o by 8%, but its performance remains below that of the Claude models. The updated results are included in the relevant sections.
> >
> > | GPT-5.1          | Exact Match | Overall Custom Score |
> > |----------------|--------------------------|---------------------------|
> > | Text           | 3.17                     | 25.51                     |
> > | 2D Image       | 0.32                     | 18.66                     |
> > | Video          | 1.00                     | 18.62                     |
> > | Backward       | 1.11                     | 20.49                     |
> > | Planning       | 2.25                     | 7.75                      |
> >
> > | Model     | Exact Match  | Overall Custom Score |
> > |------------|--------------------------|---------------------------|
> > | GPT-4o     | 0.33                     | 9.97                      |
> > | Claude 4   | 1.56                     | 19.99                     |

---

> > > ### Author Response · Authors · 2025-11-21
> > > **Response to Reviewer a6NR (Part 3)**
> > >
> > > **Dataset Size:**
> > >
> > > The automated dataset generation pipeline produces over 12,000 unique folding configurations (excluding rotation) across 16 folding types, summarized in Table 8. By **combining these configurations with randomized hole properties,9 shapes, 32 locations, 4 orientations, and 2 sizes, it can generate over 1M PFT problems.** Including rotation further expands this number.
> > >
> > > **Visual vs. Text Performance Gap:**
> > >
> > > We appreciate this insightful question, as it helps reveal performance gaps across modalities. To examine the effect of textual representation in PFT prediction, we re-evaluated model performance on 2D image and video tasks after removing direction information. Since the 2D image and text sets share identical examples, performance differences directly reflect the influence of textual input. For comparison with video-based prediction (900 problems), we selected a corresponding 315-example subset from the text-based cases.
> > >
> > > Five models, o3, Claude 4, Claude Opus 4.1, GPT-5, and GPT-4o, were evaluated across all modalities, and their results were re-analyzed without considering the direction attribute. The findings reveal that **textual representations generally yield higher exact-match accuracy than visual formats**, except for GPT-4o. Both o3 and Claude Opus 4.1 perform better at the 2D image prediction task compared to video-based evaluations. Notably, Claude Opus 4.1 shows a significant accuracy drop, decreasing from 17.46\% on text-based tasks to 4.44\% on image-based tasks. Similarly, Claude 4 demonstrates a 10\% decrease between text and image-based prediction. But both models show **comparable performance between image and video tasks.** While GPT-4o shows nearly identical performance across all modalities (0.32\%), GPT-5.1 exhibits small variations, with 3.17\% accuracy on text and 1.27\% on image and video tasks. These results indicate that the Claude models exhibit a substantial performance gap between text-based and visual evaluations.
> > >
> > > | Models           | 2D Image Exact Match | 2D Image Overall Score  | Text Exact Match | Text Overall Score | Video Exact Match  | Video Overall Score  |
> > > |------------------|--------------------------|-----------------------------|----------------------|------------------------|-----------------------|-------------------------|
> > > | o3               | 19.05                    | 39.39                       | 25.07                | 47.13                  | 15.87                 | 40.60                   |
> > > | Claude Opus 4.1  | 4.44                     | 28.53                       | 17.46                | 33.75                  | 2.86                  | 26.00                   |
> > > | Claude 4         | 2.86                     | 27.44                       | 12.38                | 30.90                  | 2.22                  | 27.18                   |
> > > | GPT-4o           | 0.32                     | 20.40                       | 0.32                 | 20.42                  | 0.32                  | 17.58                   |
> > > | GPT-5.1          | 1.27                     | 25.62                       | 3.17                 | 25.51                  | 1.27                  | 26.37                   |
> > >
> > > **Human Baseline:**
> > >
> > > Thank you for highlighting this important comparison. We agree that human evaluation is essential to understand the gap between humans and models in spatial visualization. To assess human performance, we used sample cases with up to two steps, including rotations (Groups 1, 2, and 5), and collected responses via Amazon Mechanical Turk. Each participant answered four question types: video, 2D, text-based prediction, and planning, resulting in 64 distinct questions. Since human performance was evaluated only on Groups 1, 2, and 5, we used the corresponding model results and calculated their average. The table below shows the results. In general, the models achieve higher accuracy for simpler scenarios compared to those that involve several combined folding steps. **Across different prediction modalities, human performance remains consistent, 75%; however, the models’ performance is highest for textual, then 2D, and lowest for video-based modalities.** For the planning task, humans perform similar results to the prediction task, but models struggle to create a sequence of actions with correct hole information. The evident gap in visual prediction between human participants and models indicates that current models have not yet achieved human-level spatial visualization. The analysis has been included in the paper in Appendix E.1.
> > >
> > > | Model           | Video Pred | Image Pred | Text Pred | Planning |
> > > |-----------------|-------------|-------------|------------|-----------|
> > > | Human           | 75.00       | 75.00       | 75.00      | 75.00     |
> > > | o3              | 16.66       | 21.69       | 40.56      | 14.50     |
> > > | Claude Opus 4.1 | 2.66        | 5.66        | 33.01      | 18.00     |

---

> > > > ### Comment · Reviewer_a6NR · 2025-11-27
> > > > **Follow up**
> > > >
> > > > Thank you for your response. Since the authors have addressed most of my concerns, I will raise my score to 6.
> > > >
> > > > For future revisions, I recommend adding the following:
> > > > - Human baseline details: not only overall results but also descriptive statistics, including the number of human participants and the inter-annotator correlations.
> > > > - Results from the latest state-of-the-art models, such as GPT-5.1, Gemini-3-Pro, and Claude-4.5.

---

### Official Review · Reviewer_wAZc · 2025-11-01

**Soundness:** 4
**Presentation:** 4
**Contribution:** 4
**Rating:** 8
**Confidence:** 5

**Summary:**

This paper introduces MentalBlackboard, a large-scale benchmark designed to evaluate spatial visualization ability in vision-language models through paper folding and hole-punching tasks. It offers open-ended prediction and planning tasks in text, image, and video modalities. Results show that even top proprietary models (GPT-o3, Claude Opus 4.1) perform poorly on spatial transformations involving symmetry, rotation, and sequential reasoning.

**Strengths:**

1. The benchmark’s formulation of Prediction (forward transformation) and Planning (inverse transformation) tasks—each available in video, 2D image, and symbolic-text modalities—provides a balanced framework for evaluating both causal and inverse spatial reasoning.

2. Technically rigorous dataset generation pipeline with physical validation. Used VPython to build a physically feasible 3D folding and rotation environment, automatically generating over 12,000 unique configurations (including folding, rotation, and hole-punching) to form an open-ended evaluation benchmark.

3. The evaluation dimensions are detailed: in addition to the overall accuracy, the analysis also covers four attribute dimensions: shape, size, location, and direction.

**Weaknesses:**

Overinterpretation of “Generalization” Results：
The discussion interprets the analogy-style generalization task as evidence of spatial information transfer. However, this task no longer requires mental folding or 3D transformation—only attribute mapping between similar examples. The observed success (e.g., on shape/size but not direction/location) more likely reflects linguistic or pattern-matching biases rather than genuine spatial reasoning. The authors should clarify this conceptual boundary and avoid conflating visual analogy with mental visualization.

Insufficient Analysis of Backward Folding：
The sharp performance drop on backward folds is attributed to view-dependent difficulty, but the paper provides no causal validation. It remains unclear whether the degradation stems from missing visual cues, lack of depth/layer representation, or intrinsic spatial limitations. Additional controlled ablations (e.g., providing visibility masks or depth cues) are necessary to substantiate the explanation.

Lack of Human Baseline or Cognitive Reference：
Despite frequent references to cognitive psychology and mental imagery, no human performance baseline is reported. Without such a comparison, it is difficult to gauge how far current models deviate from human-level spatial visualization. Even a small-scale human study on 10 items would provide valuable calibration.

Limited Error Analysis and Visualization：
The discussion mentions typical failure types (extra holes, missing holes, direction errors) but does not present quantitative distributions or visual examples. A systematic confusion analysis or complexity-wise breakdown would make the findings more interpretable.

Vague Future Directions.
The discussion ends with general remarks about “training models to perform mental transformations” but lacks concrete technical proposals. Potential routes such as geometry-aware modules, symmetry-consistency losses, or layer/depth supervision could be explicitly discussed to guide future research.

Overall, the Discussion section identifies important phenomena but stops short of explaining why they occur. It remains descriptive rather than diagnostic, missing an opportunity to provide causal insight and concrete improvement pathways.

**Questions:**

Missing and Unbalanced Related Work:

The section on Spatial Reasoning Benchmarks overlooks several recent and influential datasets that are directly relevant, such as VSI-Bench. In contrast, the included works (e.g., GOAT-Bench, Navi2Gaze) are tangentially related and focus on embodied or navigation-based reasoning.

The paper would benefit from a systematic taxonomy contrasting real image spatial reasoning benchmark.

Next plan:
Future versions could be extended to other cognitive dimensions, such as mental rotation, surface development, and cross-section, to form a true “multidimensional spatial intelligence” benchmark.

---

> ### Author Response · Authors · 2025-11-21
> **Response to Reviewer wAZc (Part 1)**
>
> We are encouraged by your positive feedback. We appreciate that you acknowledged MentalBlackboard as a physically grounded, large-scale, and methodologically rigorous dataset creation pipeline. Below, we provide additional clarifications and qualitative results as requested:
>
> **“Overinterpretation of “Generalization” Result"**
>
> Analogical reasoning is essential to improve the generalization ability in which the model implements the identified pattern to a new set of data. It involves the process of mapping abstract relationships between concepts. A key aspect of analogical reasoning is the transfer of knowledge from previous relational learning from one concept to another. **Transferring spatial information is not spatial reasoning; it is more mapping spatial data from one example to another, and that's why it does not require manipulation of spatial properties.** In this study, we focused on transferring spatial relationships of the hole information (direction, location, shape, and size) with visual action sequences and textual representations. The generated analogy questions consist of two cases where folding actions are the same, and only one attribute of the hole information is different in the other case. In these circumstances, models do not require spatial visualization, but calculations of hole information based on previous cases. If the generalization task includes cases where folding actions are different, then the model needs to solve the spatial visualization task. However, in the paper, instead of mental folding, we request models to apply the spatial relations are seen in the previous scenario. Although the analogy questions provide textual and visual representations of the PFT cases, the **model needs to calculate the direction and the location of the holes.** These calculations can’t be extracted from linguistic or pattern-matching biases because the requested spatial information is not the same as that provided in the previous case. For exact match accuracy, the model needs to predict all the generated hole attributes correctly. Although the accuracy of prediction and planning tasks is around 10% in visual environments, the models achieve at most 71% in generalization tasks. The best model calculates the direction and location-based analogy questions with around 55% accuracy.  This performance increase shows models can process direction and location in a simple case where a symmetric transformation is missing. This proves that the **main problem of visualization is not caused by the calculation of location and direction but by the application of mathematical transformations.**
>
> **“Insufficient Analysis of Backward Folding"**:
>
> The only difference between the videos of forward folding and backward folding prediction tasks is the direction of the folded paper. While paper is folded towards the camera/observer in forward folding, it is folded away from the camera for the backward folding process. **Once folding is completed, the visual results of both approaches are identical.** The visual example of the backward and forward folding process is included in the paper in Appendix F.
>
> **For backward folding, we utilize the same paper structure, view, and folding shadows as forward folding. The location of the camera and the zoom distance are also the same.** Therefore, the visual cues utilized in forward folding are the same as for backward folding actions. Besides, as we do in forward folding, we already provided textual captions on top of each video recording, stating the folding type and direction (backward/forward). Since both cases utilize the same action space, spatial representation of holes, and paper structure, the performance difference in partial accuracy is not caused by intrinsic spatial limitations. Besides, the final paper’s shape is always the same after the folding action no matter of the folding directions, which eliminates the lack of depth and layer representation for the reason of accuracy drop.
>
> After considering all these possible reasons within the same configuration, except the direction of the folding revealed that the degradation stems from view-dependent difficulty, which challenges models to perform the application of symmetry during the backward unfolding process. Some models, like Claude Opus 4.1, have a significant performance drop in predicting exact unfolding steps in the backward prediction case, which decreases their exact match accuracy. This result shows that applying symmetry to backward unfolding cases is more challenging for some models compared to applying it to forward unfolding cases.
>
> When we analyze both quantitative results and consider the direction as a difference in the setting, with respect to the reviewer, we do not see a reason to conduct additional ablation studies for further elaboration.  More visual examples of backward folding are included in Appendix F.

---

> > ### Author Response · Authors · 2025-11-21
> > **Response to Reviewer wAZc (Part 2)**
> >
> > **“Lack of Human Baseline or Cognitive Reference"**:
> >
> > Thank you for highlighting this important comparison. We agree that human evaluation is essential to understand the gap between humans and models in spatial visualization. To assess human performance, we used sample cases with up to two steps, including rotations (Groups 1, 2, and 5), and collected responses via Amazon Mechanical Turk. Each participant answered four question types: video, 2D, text-based prediction, and planning, resulting in 64 distinct questions. Since human performance was evaluated only on Groups 1, 2, and 5, we used the corresponding model results and calculated their average. The table below shows the results. In general, the models achieve higher accuracy for simpler scenarios compared to those that involve several combined folding steps. **Across different prediction modalities, human performance remains consistent, 75%; however, the models’ performance is highest for textual, then 2D, and lowest for video-based modalities.** For the planning task, humans perform similar results to the prediction task, but models struggle to create a sequence of actions with correct hole information. The evident gap in visual prediction between human participants and models indicates that current models have not yet achieved human-level spatial visualization. The analysis has been included in the paper in Appendix E.1.
> >
> > | Model           | Video Pred | Image Pred | Text Pred | Planning |
> > |-----------------|-------------|-------------|------------|-----------|
> > | Human           | 75.00       | 75.00       | 75.00      | 75.00     |
> > | o3              | 16.66       | 21.69       | 40.56      | 14.50     |
> > | Claude Opus 4.1 | 2.66        | 5.66        | 33.01      | 18.00     |
> >
> > **“Limited Error Analysis and Visualization"**:
> > We appreciate the reviewer’s valuable suggestion regarding the inclusion of qualitative analysis. We agree that visual examples of identified error types in both prediction and planning tasks provide a more concrete understanding of the quantitative findings. We present example **qualitative results in Appendix F**.
> >
> > The prediction task complexity is determined by the number of folds and rotations. As folding steps increase, greater sequential reasoning and visual–spatial memory are required. To assess complexity-wise accuracy, we **analyzed video prediction results across folding structures**. We selected o3 and Claude Opus 4.1 as baseline reasoning models for comparison. As the number of folds grows, o3’s exact match accuracy declines and missing holes increase, peaking at six folds with rotation. Although both error types temporarily drop in Group 5, where folding is followed by rotation, they rise again with additional steps. Unfolding precision stays around 42% up to three folds without rotation, but decreases sharply when rotations are introduced, confirming that **rotations add complexity**. In contrast to o3, Claude Opus 4.1 initially generates many extra holes (~45%) from Groups 1–5, reaching 82% at six folds. During two-step rotations, missing holes decreases, but extra holes persists. Both the exact unfold and exact match demonstrate a similar decrease as observed in o3; however, unlike o3, the model fails to solve the task that includes the two-step rotation. Detailed tables and figures are provided in Appendix E.5.
> >
> > To assess performance on one-step fold types (diagonal, vertical, and horizontal) and rotation cases, we evaluated o3 and Claude Opus 4.1 on video and text-based prediction tasks. As shown in Table 18, **o3 correctly identifies unfolding actions for horizontal and vertical folds but struggles with diagonal cases.** When rotations are added, its accuracy drops sharply from 90% to 28% for horizontal and 100% to 37% for vertical, which indicates difficulty accounting for orientation changes. **Claude Opus 4.1 performs better on diagonal folds (45%) but fails on horizontal ones** (Table 19). Although o3 correctly predicts shape and size accurately in most cases, the Claude Opus 4.1 struggles to identify them except in diagonal cases. Introducing rotation further reduces its accuracy (by 10% for diagonal and 20% for vertical) and increases the number of extra holes. The Detailed results are provided in Appendix E.5.
> >
> > The group-wise performance comparison results of the models in planning tasks are provided in Table 13 in Appendix E.2. The models require planning the folding actions based on the provided number of steps, which determines the difficulty level. **Although each scenario can be completed with a one-step process, we increase the complexity level of the task by asking the model to find the exact number of steps to generate the final paper shape.** The results in Table 13 show that when the number of steps increases, the models’ accuracy in planning the folding actions drops.

---

> > > ### Author Response · Authors · 2025-11-21
> > > **Response to Reviewer wAZc (Part 3)**
> > >
> > > **Vague Future Directions**:
> > >
> > > We thank the reviewer for this valuable suggestion. We agree that outlining concrete technical directions would strengthen the discussion. We propose a **preference-based reinforcement learning (RL) framework that aligns the model’s multi-step object manipulation with feedback from our 3D animations**. Using the detailed folding/unfolding videos in MentalBlackboard, we can leverage them as intermediate supervision targets to guide latent transformations before fine-tuning with RL. The training process is guided by fine-grained rewards that evaluate both intermediate steps and complete trajectories. The stepwise rewards include understanding the folding actions and hole information, identifying the unfolding actions in sequence, calculating the location and direction of holes during symmetry action, and increasing the awareness of the physical changes in the paper’s orientation and occlusion handling when calculating symmetry. The trajectory rewards scores whether the final prediction task succeeds in terms of the final unfolded paper and hole information. To handle increasing task complexity, we plan to apply curriculum-based training that starts with simple one-fold actions and progressively adds multiple folds and rotations.
> > >
> > > **“Descriptive Discussion Section”**
> > >
> > > Thank you for your feedback. As noted in the first question, the generalization task does not involve mental transformations since the fold types remain the same. The model only needs to detect differences between cases and apply the learned relationships from the initial folding–unfolding pairs. Since hole information of the first case is provided in both visual and textual formats, the model can leverage them for predicting the final hole results of the second case. Since the shape and size can be inferred from textual patterns, their score is higher than other attributes, location, and direction. However, the **model must compute the exact hole positions and orientations**, which cannot be inferred directly from the text. As these calculations are not required to apply symmetry, the performance of models for these two attributes is higher than for other tasks, including symmetry.
> > >
> > > As we discussed in the second question, forward and backward folding differ only in direction. Therefore, the performance difference of models between these two prediction tasks is caused by **view-dependent perception**, where some models struggle to identify unfolding actions away from the camera/observer.
> > >
> > > Hopefully, these explanations will remedy your concern about the Discussion section.
> > >
> > > **Missing and Unbalanced Related Work:**
> > >
> > > Authors agree that VSI-Bench is an influential spatial reasoning benchmark. In the related work section, we include GOAT-Bench [1] and Navi2Gaze [2] as examples of spatial orientation and navigation datasets, as both involve reasoning about location, orientation, and movement in 3D space. On the other hand, VSI-Bench focuses more on understanding space, depth, and measurement from egocentric video, which positions it more appropriately within the spatial perception and reasoning category rather than navigation. Given its importance, VSI-Bench[3] has been included in the related work section as a spatial reasoning benchmark about visual arrangements.
> > >
> > > [1] Khanna, M., Ramrakhya, R., Chhablani, G., Yenamandra, S., Gervet, T., Chang, M., Kira, Z., Chaplot, D. S., Batra, D., & Mottaghi, R. (2024). GOAT-Bench: A benchmark for multi-modal lifelong navigation. In Proceedings of the IEEE/CVF Conference on Computer Vision and Pattern Recognition (CVPR 2024)
> > >
> > > [2] https://arxiv.org/abs/2407.09053
> > >
> > > [3] https://arxiv.org/abs/2412.14171
> > >
> > > **“The paper would benefit from a systematic taxonomy contrasting real image spatial reasoning benchmark.”**
> > >
> > >  Spatial visualization tasks are different than other reasoning tasks in terms of the structure and cognitive processes. Although rotation and memory are utilized in other spatial reasoning datasets, **the main difference is the manipulation of the object in space by changing its shape**. While other spatial reasoning categories utilize real images or videos for spatial relations or spatial understanding, spatial visualization tasks require a manual 2D to 3D conversion process. Generating data with real images or videos requires extra time and effort and results in a limited number of samples. Therefore, spatial visualization benchmarks employ synthetic data for the task.
> > >
> > > **"Next plan: Future versions"**
> > >
> > > Thank you for this insightful suggestion. Thank you for the insightful suggestion. We agree that adding more spatial visualization tasks and multi-step evaluations would make the benchmark more comprehensive, and we will consider these directions in our future work.

---

> > > > ### Comment · Reviewer_wAZc · 2025-11-27
> > > >
> > > > Thanks for answering my questions! I do not have further questions and keep my rating.

---

### Author Response · Authors · 2025-11-21
**Global Response to Reviewers**

We greatly appreciate the reviewers’ thoughtful and constructive feedback. We are pleased to note their unanimous positive assessments, which highlight the strengths of our work across multiple dimensions.

- Reviewer uNrQ acknowledges that MentalBlackboard **reveals the limitations of large language and vision-language models**, supported by **detailed and comprehensive evaluations** (wAZc, a6NR, uNrQ) across attribute dimensions, folding directions, and difficulty levels.
- Reviewers wAZc and uNrQ highlight MentalBlackboard as a **balanced framework** designed to assess essential reasoning capabilities in both **causal (prediction) and inverse (planning)** contexts.
- Reviewers wAZc and uNrQ acknowledge MentalBlackboard’s significant contribution to the **rigorous and automated dataset creation pipeline**, highlighting its physical validation through VPython, **strong reproducibility, and large-scale generation** of over 12,000 unique configurations.
- Reviewer a6NR emphasizes that MentalBlackboard addresses **a clear and important gap** by evaluating the multi-step cognitive process of visualization through open-ended generation, which enables **fine-grained error analysis**.
- Reviewer a6NR highlights the generalization task as an **insightful ablation, revealing that the core limitation lies in the multi-step visualization process** itself rather than in the understanding of basic spatial properties.
- Additionally, we are pleased that reviewer uNrQ described our paper as **well-written and easy to follow**.

We have addressed each reviewer’s comments in detail. Below, we summarize our responses to the key points, highlight the corresponding changes made in the paper, and offer additional clarifications.

### **Summary of Additional Ablations**

**Qualitative Analysis**: As requested by Reviewers wAZc, a6NR, and uNrQ, we analyzed the models’ responses to generate visual examples illustrating accurate predictions, inaccuracies, and error types for each task. The results are provided in Appendix F.

**Complexity-wise Error Analysis**: As requested by Reviewer wAZc, we evaluated the complexity-wise accuracy of o3 and Claude Opus 4.1 with their video prediction results. We observed that when folding steps increases and rotation is included, the performance of the models decreases. Additionally, we observed that while o3 struggles with diagonal cases, Claude Opus 4.1 fails on horizontal scenarios.

**Speculative Failure Analysis**: In response to Reviewer uNrQ, we conducted a quantitative evaluation of specific error types in Section 5.1, using the video prediction results of o3 and Claude Opus 4.1. The results of scenarios, where a diagonal fold is followed by horizontal or vertical folds, indicate that models fail to account for layer depth when applying symmetry.

**Results of GPT-5.1**: As requested by Reviewer uNrQ, we evaluated GPT-4o and Claude 4 on the video prediction task and also tested the newly released GPT-5.1 model on all tasks. We found that GPT-4o performs equally on video and 2D images, while Claude 4 performs slightly better by 1.56%. GPT-5.1 performs slightly better than GPT-4o on video and text-based tasks.

**Visual vs Text Performance Gap**: We conducted a re-evaluation of the video and 2D image-based prediction tasks using the same dataset, omitting the direction attribute. Our findings show that textual representations achieve higher exact-match accuracy than visual representations. Both o3 and Claude Opus 4.1 perform better on the 2D image prediction task than on video-based evaluations.

**Human Performance**: In response to Reviewer wAZc and a6NR, we prepared a human performance test using Amazon Mechanical Turk. Our findings indicate that current reasoning models have not yet achieved human-level spatial visualization.

### **Changes in the Paper**:
- We have included new ablation studies in Section 6.2, Appendix E.1, E5 and F, along with tables containing quantitative results and graphs.
- In response to the comments from Reviewers wAZc and a6NR, we expanded the Related Work section and Table 1 by adding three additional benchmarks to strengthen the contextual background of our study.
- In response to the comments from Reviewer a6NR, we added a clarification of the unfolding action space in Section 3.1. We also revised Tables 2, 13, 14, 16 and 17, as well as Figures 5 and 7, to include the experimental results of GPT-5.1.
- In response to Reviewer uNrQ’s comments, we revised the placement of the figures to better align them with the corresponding text and added color to Table 2 to improve clarity.
- In response to Reviewer wAZc’s comments, we expanded Section 6.3 by adding further explanations to clarify the discussed concepts.

Once again, we thank the reviewers and the Area Chairs for their time and detailed feedback. We hope that our responses have addressed all remaining concerns and questions. We look forward to further discussion if needed.

---

> ### Author Response · Authors · 2025-12-01
> **Revised Scores**
>
> We thank the reviewers and the AC for their time and constructive feedback. For reference, initial scores were **4, 6, and 8**, which, after our responses and additional experiments, were updated to **6, 6, and 8**.

---

### Note · Program_Chairs · 2026-01-17
**Submission Desk Rejected by Program Chairs**

The following references in this submission do not refer to real documents and/or have major errors in bibliographic information:

 Geoff Woolcott. A multi-level conceptual framework for integrated STEM education. International Journal of STEM Education, 7(1):11, 2020. doi: 10.1186/s40594-020-00210-1. URL https://doi.org/10.1186/ s40594-020-00210-